# Improving the workflow to crack Small, Unbalanced, Noisy, but Genuine (SUNG) datasets in bioacoustics: The case of bonobo calls

**Vincent Arnaud**[1,2], **François Pellegrino**[2], **Sumir Keenan**[3], **Xavier St-Gelais**[1], **Nicolas Mathevon**[3], **Florence Levréro**[3], **Christophe Coupé**[2,4] *

**1** Département des arts, des lettres et du langage, Université du Québec à Chicoutimi, Chicoutimi, Canada, **2** Laboratoire Dynamique Du Langage, UMR 5596, Université de Lyon, CNRS, Lyon, France, **3** ENES Bioacoustics Research Laboratory, University of Saint Étienne, CRNL, CNRS UMR 5292, Inserm UMR_S 1028, Saint-Étienne, France, **4** Department of Linguistics, The University of Hong Kong, Hong Kong, China

* ccoupe@hku.hk

**Data Availability Statement:** All relevant data are within the manuscript and its Supporting

## Abstract

Despite the accumulation of data and studies, deciphering animal vocal communication remains challenging. In most cases, researchers must deal with the sparse recordings composing Small, Unbalanced, Noisy, but Genuine (SUNG) datasets. SUNG datasets are characterized by a limited number of recordings, most often noisy, and unbalanced in number between the individuals or categories of vocalizations. SUNG datasets therefore offer a valuable but inevitably distorted vision of communication systems. Adopting the best practices in their analysis is essential to effectively extract the available information and draw reliable conclusions. Here we show that the most recent advances in machine learning applied to a SUNG dataset succeed in unraveling the complex vocal repertoire of the bonobo, and we propose a workflow that can be effective with other animal species. We implement acoustic parameterization in three feature spaces and run a Supervised Uniform Manifold Approximation and Projection (S-UMAP) to evaluate how call types and individual signatures cluster in the bonobo acoustic space. We then implement three classification algorithms (Support Vector Machine, xgboost, neural networks) and their combination to explore the structure and variability of bonobo calls, as well as the robustness of the individual signature they encode. We underscore how classification performance is affected by the feature set and identify the most informative features. In addition, we highlight the need to address data leakage in the evaluation of classification performance to avoid misleading interpretations. Our results lead to identifying several practical approaches that are generalizable to any other animal communication system. To improve the reliability and replicability of vocal communication studies with SUNG datasets, we thus recommend: i) comparing several acoustic parameterizations; ii) visualizing the dataset with supervised UMAP to examine the species acoustic space; iii) adopting Support Vector Machines as the baseline classification approach; iv) explicitly evaluating data leakage and possibly implementing a mitigation strategy.

Information files. Code is available at http://github.com/keruiduo/SupplMatBonobos.

**Funding:** We thank the Ministère de l'Enseignement Supérieur et de la Recherche (France) (https://www.enseignementsup-recherche.gouv.fr/en) and the Ecole Doctorale SIS of the University of Saint-Etienne (https://edsis.universite-lyon.fr/) (Ph.D. grant to SK, n° ENS 2012/398), the Université du Québec à Chicoutimi (http://www.uqac.ca) (VA), the University of Saint-Etienne (https://www.univ-st-etienne.fr/fr/index.html) (research sabbaticals to FL and NM, visiting professorship to VA and research grant), the LABEX ASLAN (ANR-10-LABX-0081) (https://aslan.universite-lyon.fr/) (FL, VA, and FP) of the University of Lyon within the Investments for the Future program (ANR-11-IDEX-0007) operated by the French National Research Agency (https://anr.fr/), the Institut Universitaire de France (https://www.iufrance.fr/) (NM), and The Canada Graduate Scholarship (XSG) – Michael Smith Foreign Study Supplements (CGS-MSFSS) of the Social Sciences and Humanities Research Council of Canada (SSHRC) (https://www.sshrc-crsh.gc.ca/home-accueil-eng.aspx). The funders had no role in study design, data collection and analysis, decision to publish, or preparation of the manuscript.

**Competing interests:** The authors have declared that no competing interests exist.

## Author summary

Deciphering animal vocal communication is a great challenge in most species. Audio recordings of vocal interactions help to understand what animals are saying to whom and when, but scientists are often faced with data collections characterized by a limited number of recordings, mostly noisy, and unbalanced in numbers between individuals or vocalization categories. Such datasets are far from perfect, but they are our best chance to understand communication in hard-to-record species. Opportunities may especially be limited to record endangered species such as our closest relatives, bonobos and chimpanzees. We propose an efficient workflow to analyze such imperfect datasets using recent methods developed in machine learning. We detail how this approach works and its performance in unraveling the complex vocal repertoire of the bonobo. Our results lead to the identification of several practical approaches that are generalizable to other animal communication systems. Finally, we make methodological recommendations to improve the reliability and reproducibility of vocal communication studies with these imperfect datasets that we call SUNG (Small, Unbalanced, Noisy, but Genuine datasets).

## Introduction

Cracking animal vocal communication codes has always constituted a motivating challenge for bioacousticians, evolutionary biologists and ethologists, and impressive breakthroughs have been achieved in the last decade thanks to advances in both data collection and machine learning, in conjunction with well-designed experimental approaches (e.g., [1–3]). Building upon these remarkable achievements, can we conjecture that the puzzle of animal vocal communications will be soon solved? Probably not really. The high level of understanding achieved for a few dozens of species should not obscure the fact that for the vast majority of animal species, not much is known yet [4]. For most animal species, neat, clean, and massive datasets are out of reach and bioacousticians have to cope with Small, Unbalanced, Noisy, but Genuine (SUNG) datasets characterized by data paucity, unbalanced number of recordings across individuals, contexts, or call categories for instance, and by noisy and sometimes reverberant recording environments. Despite their imperfection, such datasets, which are often time-consuming to collect and require extensive and specific expertise to annotate, are precious as they inform us about the complex communication systems of animal species that may be dramatically endangered or of major scientific interest. As a result, it is becoming increasingly important to adopt the best possible practices in the analysis of such datasets, both to provide reproducible studies and to reach robust and reliable conclusions about the species communication systems, and beyond that, about more general questions concerning the evolution, diversity, and complexity of communication systems. We propose in this paper an operational workflow designed to help bioacousticians solve problems commonly encountered in concrete situations with SUNG datasets and we illustrate its relevance in a case study addressing the bonobo (*Pan paniscus*) vocal repertoire.

In mammals, individuals often produce vocalizations potentially informative to their conspecifics in their "here and now" context of emission. Additionally, these signals can also provide idiosyncratic clues to the emitter identity, which is often an essential information for territorial defense, social interaction and cohesion (e.g., [5]). In social species, especially those living in fission-fusion systems (i.e., the size and composition of the social group change over time with animals splitting (fission) or merging (fusion) into subgroups), the "who" is

therefore as important as the "what" in vocal communication (e.g., [5,6]). Much research has therefore sought to determine which acoustic primitives (a.k.a. features) encode the "who" and the "what" respectively in order to test hypotheses about the functions fulfilled by vocal communication, through playback experiments of both natural and resynthesized sound stimuli for instance [7–9]. In some communication systems, the emitter identity and the contextual information are sequentially encoded (a strategy called temporal segregation), leading to a straightforward identification of their respective acoustic primitives [10], but in other species, distinguishing the acoustic features on which the context-specific information and the vocal signature develop can be much more complex and challenging [11,12].

A key step to disentangle the "who" from the "what" is thus to assess the discriminative power of potential acoustic *features* by automatic classification in order to infer their putative role in communication as signals, carrying some information about the identity or intent of the emitter, the call type, and the context of utterance. This process belongs to the field of supervised machine learning and consists in training a classifier to discriminate data samples (training set) according to a priori categories (labels). The performance of the classifier is then measured in terms of its ability to correctly generalize the discrimination decision to new unseen data samples (also described as observations) belonging to the same categories (test set). A common–but sometimes overlooked–detrimental issue occurs when the classifier's decision is correct, but based on faulty premises because the samples in the training and test sets are not drawn from independent datasets and share confounding properties other than the category label itself (e.g., a background acoustic noise that leaks information about the identity of the recorded individual), a phenomenon known as data leakage in data mining and machine learning ([13]; see also the "Husky vs Wolf" experiment in [14], for a striking illustration). Fifteen years ago, Mundry and Sommer showed how permutation tests–a well-known category of non-parametric tests in statistics [15,16]–could be combined with Discriminant Function Analysis (DFA) to limit the risk that such non-independence would lead to an overestimation of discriminant power [17]. In substance, permuted DFA (pDFA) performs a robust estimation of discriminability by comparing the correct classification performance reached with the real data to the classification distribution resulting from a large number of random permuted versions derived from this real dataset. For instance, if the performance reached with the actual data falls into the top 1% of the distribution, a significant discriminability is acknowledged (with alpha = 1%). This procedure results in a quite conservative and accurate evaluation of the genuine discriminability ascribable to the differences in categories compared to confounds due to potential data leakage. It has become the standard approach for bioacoustic analysis and it is still routinely used nowadays despite its limitations (e.g. [18,19] in chimpanzees; [20] in Dwarf mongoose; [21] in woodpeckers, among recent publications; see also [22] in zebra finch for a comparison with two other classifiers). Being derived from classical DFA, it indeed shares its main shortcomings: it is quite sensitive to the presence of outliers in the dataset and the maximal number of features as well as the number of observations that can be considered are quite constrained by the dataset structure and the dependency among the observations (see [17] for a thorough discussion). Moreover, it is neither the best nor the most accurate classification algorithm for assessing discriminability with SUNG datasets (see below for alternatives). As a paradoxical consequence, pDFA can be expected, on the one hand, to underestimate the information present in a dataset (because of suboptimal classification), and, on the other hand, to overestimate the class discriminability (because of residual non-independence). Meanwhile, in other scientific fields, impressive improvements have been made to address similar issues, by implementing more powerful statistical and machine-learning algorithms in more controlled configurations of evaluation. As aforementioned, such algorithms, including deep-learning neural networks, have recently been successfully applied to

animal communication, mainly in situations where data paucity was not an issue (e.g., [23], but see [2, 24] for applications to smaller datasets; [25] for a recent review; [26] for a broader perspective).

In addition to automatic classification, graphical exploration of a corpus of animal calls projected into an informative feature space is often an essential step in understanding the structure of their repertoire. In the "discovery phase", characterized by the absence of pre-existing labels assigned to each call (a so-called unsupervised situation), such a graphical representation can suggest the existence of underlying classes. At a later stage, when labels on the call type or the individual emitter have already been assigned by human experts, such labels can be used to guide the graphical projection (supervised situation). The resulting representation thus helps diagnose the consistency and adequacy of manual labeling. Since multiple acoustic features are usually involved, a reduction in dimensionality is required to get a human-friendly low-dimensional representation, usually in two (plane) or three (volume) dimensions. While this reduction was traditionally achieved through linear or near-linear transformations, such as Principal Component Analysis or Multidimensional Scaling, innovative non-linear approaches such as t-distributed stochastic neighborhood embedding (t-SNE, [27]) and Uniform Manifold Approximation and Projection (UMAP, [28]) have recently emerged. These methods generally result in intuitive representations of the local structure present in complex datasets, at the expense of the significance of the global structure. Both t-SNE and UMAP have already been successfully applied to animal communication, either as exploratory methods to assess repertoire discreteness vs. grading, or to compare the relevance of several feature sets (see [29–32] for examples; [33,34] for thorough discussions of the potential benefits of these methods, as well as their limits with small datasets).

In this paper, we apply an automatic classification workflow to a SUNG dataset whose structure should seem fairly conventional to most bioacousticians: several recordings of calls produced in sequences of varying lengths, belonging to half a dozen types, and produced by a dozen individuals. This dataset consists of audio recordings of calls emitted by captive bonobos (*Pan paniscus*). It provides a case study where the identification of individuals on the basis of their vocalizations and the identification of call types are not trivial. The bonobo vocal repertoire was described several decades ago in two seminal studies that highlighted its graded nature [35,36]. It is structured on almost a dozen prototypical types that conjugate modulated voiced vocalizations with scream components, also exhibiting nonlinear phenomena. Although quantitative studies with bonobos are still rare, it has recently been shown that an individual signature is detectable in bonobo vocalizations and that the reliability of this signature differs from one call type to another [37]. Here we implement a systematic comparison of several classification approaches to assess the strength and stability of individual bonobo signatures in the vocal repertoire. Our results establish whether the level of performance is the result of a lack of intrinsically encoded information in the vocalizations or a suboptimal classification approach.

The research question we have addressed is therefore can state-of-the-art automatic classification approaches lead to a more accurate estimation of the information encoded in a SUNG dataset and to a more comprehensive understanding of the functioning of an animal communication system than a DFA approach? We present and evaluate the relevance of several methods that can be used to overcome the difficulties inherent in SUNG datasets. In the end, the objective of this paper is to propose an operational workflow that can be used by bioacousticians in concrete situations.

Fig 1 gives a graphical overview of the proposed methodology and mirrors the paper organization. More specifically, in Section I (Fig 1; block A), we first introduce the main characteristics of SUNG datasets and we present the repertoire of bonobo calls along with the most

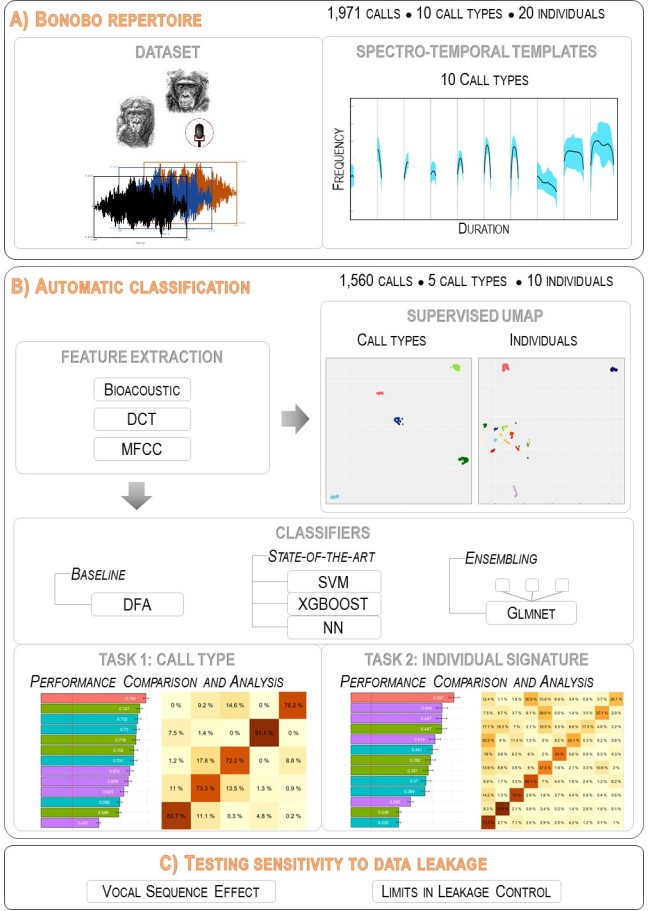

**Fig 1. Workflow implemented to analyze a dataset of animal vocalizations.** Block A is species-dependent and illustrates the bonobo case. The other blocks are generic over SUNG datasets. A. The traditional bioacoustic approach is applied to the bonobo dataset to deduce call type templates. B. Three different sets of acoustic features are associated (BIOACOUSTIC, DCT, and MFCC) to characterize the bonobo acoustic space. Supervised UMAP is run to visually assess call type and individual separability. The performance of three state-of-the-art classifiers and their ensembling combinations is assessed and compared to that of a discriminant analysis (DFA) in two tasks: identification of call types (bonobos have a vocal repertoire composed of different calls) and discrimination between emitters (identification of individual vocal signatures). C. The sensitivity of accuracy to the composition of the training and test sets and to the induced data leakage is then evaluated.

salient aspects of its graded structure and individual variability as revealed by a typical bioacoustic quantitative approach. These repertoire and dataset descriptions are by definition quite species-dependent but the general principle in generic over animal communication studies.

In Section II (Fig 1; block B), we introduce two feature sets (namely DCT and MFCC sets) in addition to the standard bioacoustic one to improve the robustness of the analysis of noisy vocalizations. A S-UMAP representation leads to a visual confirmation of the bonobo repertoire structure and to a first evaluation of the difficulty of the automatic classification tasks (call type classification and individual signature detection). We then test three classification algorithms that are compatible with the limited amount of data available and its imbalance between categories, and we report results according to several metrics deemed appropriate for SUNG datasets. We also build combined/stacked classifiers to test the complementarity between parameter sets and classification algorithms, with the aim of improving the overall robustness of the analysis compared to a baseline obtained by a DFA approach classically used

in bioacoustics. Both the graphical representation and the automatic classification approaches developed in this section are highly generic and are transposable to other SUNG datasets.

In Section III (Fig 1 block C), we provide an in-depth evaluation of the impact of data leakage due to non-independence on classification results. We first demonstrate why data leakage should not be overlooked by reporting performances reached when increasing the independence between the training and test datasets. We compare them to results from fully random training / test partitions and from (purposefully ill-designed) partitions increasing dependence. We emphasize that fixing potential data leakage issues is an important matter, and show how the use of a genetic algorithm allows the construction of training and test datasets which generate less biased accuracy estimations. We also illustrate how challenging it is to completely avoid data leakage with SUNG datasets.

Section IV discusses the main achievements and limitations of our study, and presents our final recommendations for dealing with SUNG datasets.

In Section V, we detail the methodology and discuss how bioacousticians can apply this data science approach to explore biological questions. While key methodological aspects are also introduced throughout the paper for ease of understanding, most of the details are only discussed in this comprehensive section in order to preserve legibility.

Datasets and analysis codes are made available on the online repository http://github.com/keruiduo/SupplMatBonobos as RMarkdown html pages. In addition, a step-by-step demonstration of the streamlined workflow we recommend in Section V is also provided.

## I. A SUNG dataset: The bonobo recordings

### What is a SUNG dataset?

A SUNG dataset is characterized by data sparsity and paucity, imbalance in the number of recordings between individuals, contexts or call types, and noisy and sometimes reverberant recording environments. A first set of these constraints is inherent in field conditions. Whether in zoos or in the wild, the recordings of vocalizations often involve few individuals (usually less than two dozen), in an unbalanced proportion of call types, contexts, and individuals.

**Small.**   In some rather rare situations, large corpora comprising dozens of thousands of calls can be gathered (e.g., [38–40]) and sometimes even labeled in terms of individual emitter or vocalization type [41,42]. In most cases, however, bioacousticians must deal with a limited dataset that would be deemed small in the computer and data science community. Indeed, a dataset consisting of a few thousand observations is considered small in these domains (e.g., [43]). Therefore, many studies in bioacoustics (and in many other scientific fields) are based on quantitative analyses performed on small datasets that could hardly benefit from standard machine learning approaches.

**Unbalanced.**   Datasets often present an imbalance between the categories that one would like to automatically characterize. This can be observed in large datasets ([44] in terms of call categories), but it is even more common when the dataset size itself is moderate. It can then involve vocalization types, contexts or individuals [45] and even species in monitoring applications [43,46].

**Noisy.**   Recordings are always performed in environments with unique characteristics in terms of background soundscape and noise. All these aspects impact the extraction of acoustic features and automatic classification, limiting the quantity and quality of available data and exposing the evaluation to potential misinterpretation due to data leakage. These obstacles are well identified in ecology and ethology studies (see e.g., [47] for an elegant proposal on acoustic feature extraction and modeling in the context of bird identification), but they remain

problematic, in stark contrast to human language studies for which massive data are now available (see e.g., [48] for a recent comparison of automatic speech recognition systems).

**Genuine.** Although field-recorded datasets are not perfect, they provide a distorted but genuine view of a species' communication system. Sources of distortion can be that some individuals or vocalization contexts are more difficult than others to record (impacting the number of observations) or that some vocalizations occur in a more degraded acoustic environment (impacting the quality of the observations). Such datasets, which are nevertheless made up of real observations, are insightful to characterize a communication system and its variability.

Tackling a SUNG corpus thus requires overcoming these limitations. Although there is no magic recipe, in the rest of the paper we will build upon the case of the bonobo to showcase a quite generic procedure that can be adopted for other animal communication systems. Bonobos–and great apes in general–pose a hard problem to bioacousticians. All species (*Gorilla beringei*, *Gorilla gorilla*, *Pan paniscus*, *Pan troglodytes*, *Pongo pygmaeus*) are endangered or critically endangered [49]. Collecting data in the field or in captivity is thus challenging and limited to few accessible groups of animals, leading to imperfect datasets that combine most of the concerning aspects of SUNG datasets. In substance, bonobo vocalization datasets are too small, too noisy, and involve too few animals (in a word sub-optimal) compared to corpora that can be routinely–or at least more easily–recorded in other species. As such, it provides an exemplary case to assess the procedure we propose for SUNG datasets, although some aspects remain specific to the bonobo case (such as the details of the acoustic parametrization).

## A quick overview of bonobo vocal communication

Bonobos use to vocalize in much of their daily activities and different call types are used in a flexible way across contexts, resulting in complex and meaningful combinations (e.g., inter-party travel recruitment; food preference; see [36,50,51] among others). The bonobo vocal repertoire is complex and graded: acoustic characteristics can vary within each call type, and the different types are distributed along an acoustic continuum. Two descriptions of call types have been proposed and are largely converging [35,36]. Most calls are confidently labeled by experts as belonging to one of 12 types (peep, yelp, hiccup, peep yelp, soft bark, bark, scream bark, whining whistle, whistle, scream whistle, grunt and scream), despite the gradual changes that lead to some degree of uncertainty. However, doing this automatically is a much more difficult challenge than in other primate species with a discrete repertoire (e.g., common marmosets [2,52]). This classification is based on the characteristics of the tonal contour when detectable (frequency modulation and duration) and the vocal effort (illustrated by the presence of energy in the higher harmonics and nonlinear phenomena).

Besides the "what" contextual information, the "who" information is crucial to navigate the complex fission-fusion society of bonobos. Recent research suggests, however, that the individual vocal signature is more salient in high-arousal calls than in low-arousal calls [37]. According to this result, the identification of an individual would therefore be easier on the basis of high-arousal calls. However, playback experiments have shown that bonobos are able to discriminate a familiar from an unfamiliar congener based on low-arousal calls (peep-yelp; [53]). The propensity for a call type to encode individual variation may actually vary across a repertoire. This kind of situation is not surprising per se since it had already been reported in other mammal species [54,55].

Other limitations arise from the fact that the bonobo vocal repertoire is graded and only partially understood. Thus, even tagging a bonobo call is a more complex task due to an absence of clear acoustic boundaries between call types inherent to a graded repertoire, as opposed to an ungraded or less graded repertoire (e.g., [2]). This leads to a difficulty in

identifying occurrences belonging to a potential ground truth (or gold standard) against which the performance of automatic classification approaches could be evaluated. This gold standard problem disappears when automatically identifying the call emitter, whose correct identity is known when the action has been directly observed in absence of overlaps between emitters. But even this supposedly easy situation can be complex in bonobos, as their vocal activity is often unpredictable, making the detection of the emitter difficult among all group members. In addition, a first emitter often triggers vocalizations from a few other individuals, resulting in a sudden intense vocal activity (with overlapping vocalizations). In such situations, unambiguous assignment of an emitter identity to a call produced in a sequence is sometimes impossible, and it requires a lot of recording time to obtain enough calls for which the emitter identity is unambiguously known. Moreover, even if the emitter's identity is known, deciding whether an acoustic feature is relevant or not in identifying the emitter is not straightforward. To illustrate, let's imagine that the acoustic feature A allows an automatic classifier to perfectly identify one individual when it is the emitter, but poorly performs in recognizing all other individuals. On the contrary, considering only feature B leads to a slightly better-than-chance identification for all individuals. Which feature is the most important from the animal's point of view? In this case, the answer is probably both, but this example aims to highlight that the choice of the right evaluation metrics (average recognition rate, accuracy, etc.) among the dozens available may influence the resulting ethological interpretation (see 'Performance Metrics' in Section V).

In this study, we use a dataset from the corpus analyzed in [37]. It comes from 20 adult bonobos (14 females, 6 males) housed in three zoos (Apenheul in The Netherlands, La Vallée des Singes in France and Planckendael in Belgium), totaling 380 hours of recording. Recordings were made during daytime using a Sennheiser MKH70-1 ultra-directional microphone and a Zoom H4 Digital Multitrack Recorder (44.1 kHz sample rate, 16 bits per sample, .wav files), in various contexts (foraging, grooming, aggression, etc. See [37] for details). A typical recording session includes periods with low vocal activity and others with numerous vocalizations in sequences that often intertwine several individuals, as explained above. As a consequence, the dataset consists of recordings done across quite a long period of time, but also of calls produced within the same vocal sequence of a recording session. This aspect is important when addressing the data leakage issue (Section III). Vocalizations were manually segmented, identified and then double-checked by two other experimenters (through a consensus decision-making) based on a visual inspection of the signals on spectrograms and an estimation of the fundamental frequency, $f_0$, using the speech analysis software Praat [56].

The audio quality of the recordings was variable, with many calls being recorded in a reverberant and challenging environment for automatic $f_0$ detection, often leading to uncertainties in Praat. The temporal modulation of $f_0$ was thus derived semi-automatically from the narrow-band spectrograms, thanks to a homemade Praat script based on mouse input of at least two points on the $f_0$ trace on the spectrogram by the experimenter, allowing an interpolated trajectory to be estimated. Samples of spectrograms for the five most common call types are given in Supplementary Information (S1 Fig).

Fig 2 illustrates why this dataset can be qualified as SUNG: the recorded environment may be a distant free-ranging enclosure or an indoor room; there is a quite noticeable imbalance in the number of calls per individual (see details in Table 1), and the audio bouts are noisy, reverberant, and complex to analyze.

We worked with a dataset consisting of 1,971 calls from 20 subjects to perform the preliminary quantitative study to characterize the tonal contour of each type (Section II and Fig 2) (after removing grunts and screams that do not have any tonal component). The following 10 call types are thus described: Peep (P), Yelp (Y), Hiccup (H), Peep Yelp (PY), Soft Bark (SB),

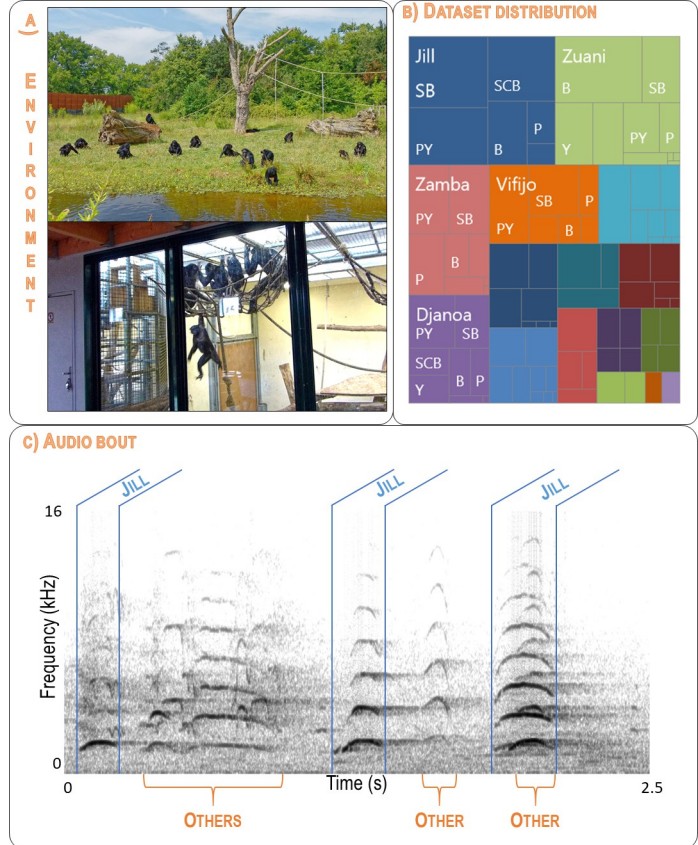

**Fig 2. An example of a SUNG bioacoustic dataset: recordings of bonobo calls in social contexts.** A. Each individual can be recorded in outdoor enclosures and inside buildings. B. The number of calls varies between individuals (unbalanced distribution coded by colored rectangles) and call types (coded by internal rectangles for each individual). The five most-represented individuals are named. The four least represented individuals are not shown on the chart. The detailed breakdown is given in Table 1. C. Spectrogram of a typical recorded bout (2.5 seconds extracted from the Jill698 recording) showing the difficulty of isolating good quality calls. A sequence of three Soft Barks produced by Jill can be identified (sections delimited by blue boundaries). Other individuals vocalize in the background (sections marked with orange curly brackets). Jill's third call is not analyzed as it overlaps too much with other vocalizations. The Jil698 recording is available as S1 Sound and described in S4 Text. Photo credits: F. Levréro (Top) & F. Pellegrino (Bottom).

Bark (B), Scream Bark (SCB), Whining Whistle (WW), Whistle (W), and Scream Whistle (SCW). For the automatic classification tasks reported in Sections II and III, we selected the ten individuals (7 females, 3 males) for whom at least 70 calls were available from the five most

**Table 1. Number of calls per individual and per call type in the dataset used for automatic classification.** The five call types are: Bark (B), Peep (P), Peep Yelp (PY), Soft Bark (SB), and Scream Bark (SCB).

| Types | Individuals | | | | | | | | | | Total |
|---|---|---|---|---|---|---|---|---|---|---|---|
| | Bolombo | Busira | Djanoa | Hortense | Jill | Kumbuka | Lina | Vifijo | Zamba | Zuani | |
| B | 6 | 11 | 20 | 9 | 50 | 23 | 8 | 12 | 21 | 114 | 274 |
| P | 27 | 18 | 18 | 26 | 24 | 35 | 24 | 20 | 43 | 20 | 255 |
| PY | 24 | 34 | 49 | 50 | 89 | 26 | 18 | 61 | 56 | 36 | 443 |
| SB | 17 | 5 | 36 | 8 | 112 | 25 | 23 | 51 | 55 | 50 | 382 |
| SCB | 2 | 3 | 22 | 18 | 87 | 0 | 3 | 16 | 18 | 37 | 206 |
| Total | 76 | 71 | 145 | 111 | 362 | 109 | 76 | 160 | 193 | 257 | 1,560 |

frequent call types Bark (B), Soft Bark (SB), Peep (P), Peep Yelp (PY) and Scream Bark (SCB) (1,560 calls, split by call types and individuals in Table 1).

### A sketch of across-individual variability

For each call type exhibiting a tonal contour (all but grunts and screams), we estimated a template (or prototype) of the fundamental frequency ($f_0$) corresponding to the average $f_0$ trajectory estimated over all its samples. $f_0$ contour extraction was automatically performed in Praat and manually corrected for coarse errors (typically subharmonic detection). These $f_0$ templates are presented in Fig 3 for the ten types of tonal call for which $f_0$ can be extracted. It should be noted that the Hiccup (H), Whining Whistle (WW) and Scream Whistle (SCW) are very rare vocal productions and their templates thus were estimated from a small number of samples, which led us not to consider them in the following sections. Considering only the bell curve types (P, PY, SB, B, and SCB), a continuum is visible on the $f_0$ dimension with increasing $f_0$ average value and excursion, except between barks and scream barks, which differ mainly in the absence or presence of a screaming component (like deterministic chaos, see [57,58]). This aspect is not captured by the $f_0$ trajectory, but it is suggested by the rather large difference in average harmonicity (see 'Extraction of acoustic features' in Section V for methodological information) between the two categories, with SCB's harmonicity being on average 2.2 dB lower than that of B. SCB thus has the lowest harmonicity among the ten types displayed in its energy distribution, leading to a highly salient perceptual roughness.

To further illustrate the graded aspect of the bonobo vocal repertoire, we also computed $f_0$ templates at the individual level. Their distribution is shown in Fig 4 (left), with a miniature of each template (at scale $1/10^{th}$) represented in a two-dimension space: average $f_0$ and average duration. A large variation in $f_0$ is observed among the individual peep templates, and their short duration distinguishes them from the other types. The Bark type spans a large area of the acoustic space, with a large variation in both duration and $f_0$. For a given individual, the relative weight of the temporal and frequential dimensions may differ, as illustrated in Fig 4 (right) for individuals #19 and #20. On average, the calls produced by individual #20 are higher-pitched than those produced by individual #19, but an additional difference is highlighted by the respective position of their SB templates which is very close to B for #19 while it is more intermediate between Yelp and Bark in terms of duration for #20. This observation suggests that the inter-individual variation observed is not entirely constrained by anatomical differences and that each individual's repertoire is akin to an idiolect in human linguistics. This type of graded repertoire with overlapping categories represents a difficult challenge for automatic classification methods [59].

## II. Automatic classification: From DFA to state-of-the-art approaches

### Extraction of acoustic features

For a given species, the task of choosing a feature space adapted to its vocalizations is not trivial given the broad diversity of production mechanisms [60]. Ethological and bioacoustic expertises are thus required to identify an initial set of relevant dimensions (duration, spectral energy distribution, etc.) and to refine it further. This leads to species-specific feature sets, such as the **Bioacoustic** set used here in the context of the bonobo SUNG dataset, following a fairly standard approach on primate call analysis, which has already been used in bonobo studies (e.g., [50]).

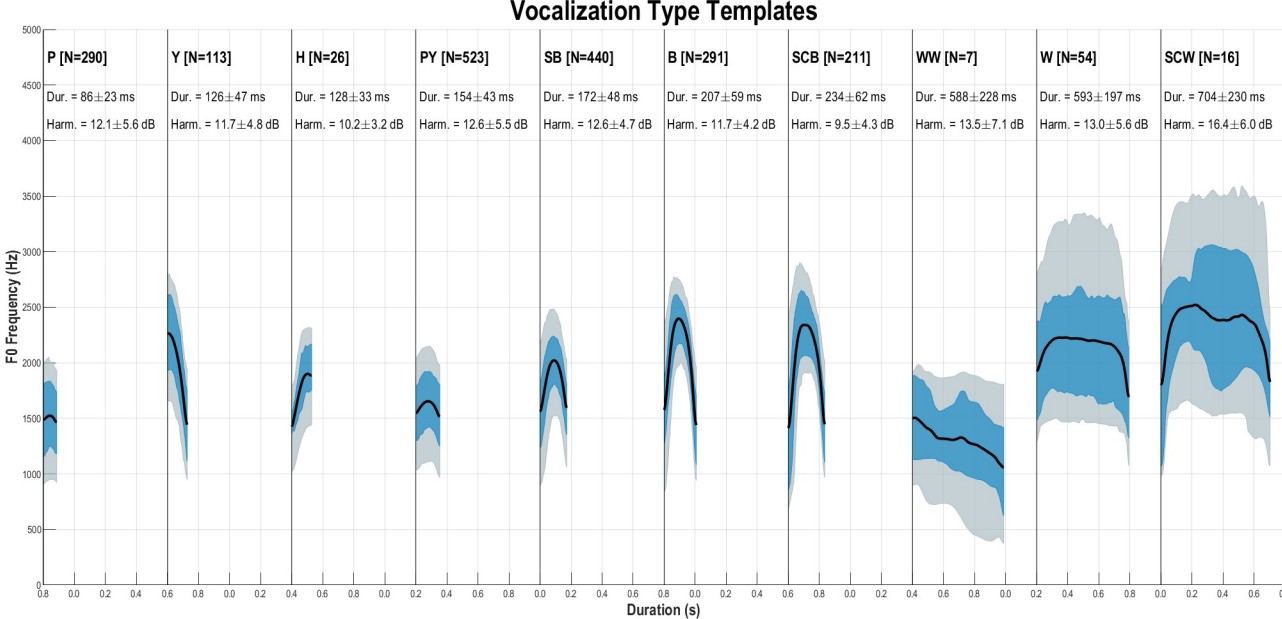

**Fig 3. Templates of $f_0$ (pitch) for each call type.** The average $f_0$ trajectory (black line) is calculated from all recordings (using Praat). The shaded area covers 50% or 80% of the distribution (blue and grey areas respectively). For each type of call, individual calls were time-scaled to the average duration of the type. N = number of calls analyzed; Dur = call duration (mean and standard deviation, in ms); Harm = harmonicity (mean and standard deviation, in dB). The types are ranked by increasing average duration: Peep (P), Yelp (Y), Hiccup (H), Peep Yelp (PY), Soft Bark (SB), Bark (B), Scream Bark (SCB), Whining Whistle (WW), Whistle (W), and Scream Whistle (SCW).

To investigate each individual vocalization, we additionally considered two sets of features: the MFCC set and the DCT set. All three feature sets are detailed in Section V and summarized in Supplementary Information (S1 Table). The **MFCC** set is adapted from the Mel-Frequency Cepstral Coefficient analysis routinely used in human speech processing [61]. Although less common, it has already been successfully applied to primate call recognition and individual identification (e.g., [3,62]). This full-spectrum approach is agnostic in the sense that it does not target specific call characteristics (such as rising or falling pitch, harmonic structure, etc.) but it may potentially be able to highlight fine-grained spectral differences that are not captured with the standard bioacoustic approach. To our knowledge, the **DCT** set has never been used for studies of primate vocalizations. It is based on studies of human speech where DCT (Discrete Cosine Transform) coefficients are useful for characterizing time-varying sounds, such as diphthongs, in terms of additive cosine components [63,64]. With only 7 dimensions, it is a minimal set characterizing the tonal contour of the call (its $f_0$ trajectory), its acoustic roughness (approximated by its harmonics-to-noise ratio), and its duration. Adopting these three feature sets is intended to test which are the most efficient in a classification task, whether they are redundant or complementary, once feature correlation is accounted for by the classification procedure (see Section V for details).

## A graphical assessment: Supervised UMAP

The dataset includes 1560 rows and 217 features, the target variables and additional metadata. All features are numeric and to perform exploratory data analysis (EDA) on this dataset, we took advantage of UMAP in its supervised version (S-UMAP). It provides an informative graphical representation useful to a) check the adequacy between the manual labeling of data points and their features (in our case, the labels pertain to call types and individual signatures),

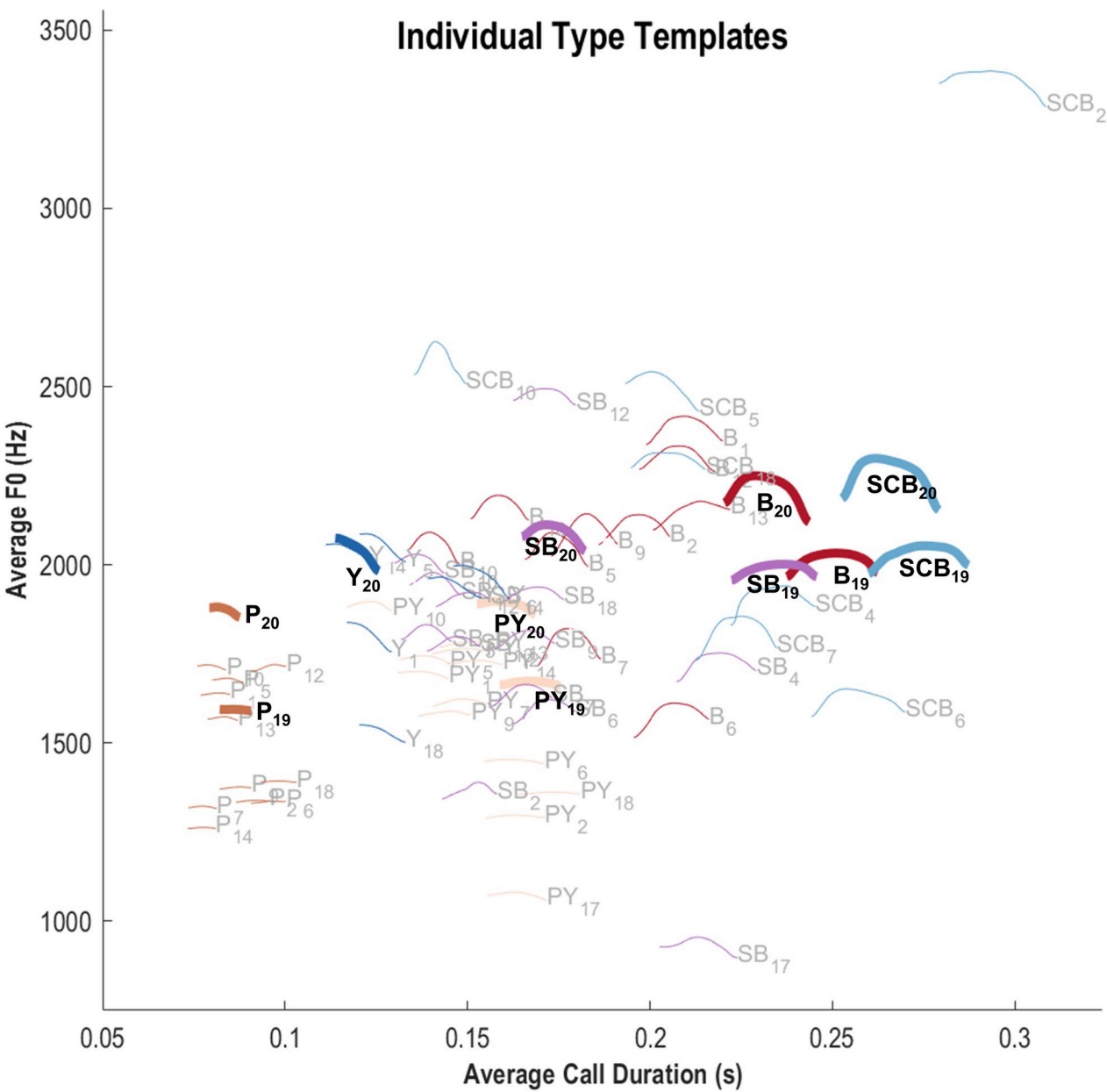

**Fig 4. Representation of call $f_0$ templates at the individual level.** Each color/hue combination corresponds to a call type (P, Y, PY, SB, B, SCB, as defined in Fig 3). Each curve is a miniature of an individual's $f_0$ template. The call type (acronym and color) and individual identity (numerical index) are indicated. All individuals and call types for which at least 3 samples were available are displayed. The repertoire of individuals #19 and #20 is highlighted (thick lines).

b) check the presence of extreme data that could indicate an erroneous label, and c) assess the degree of clustering of the different groups and thus estimate the difficulty of the task of automatic classification.

In order to estimate the quality of the partitioning provided by S-UMAP and thus quantify the degree of clustering of call types and individual signatures, we computed the values of the silhouette scores [65]. Silhouette scores (see also Section V) were originally developed as a graphical aid for the interpretation and validation of cluster analysis. They provide a measure

of clustering quality based on both the cohesiveness of the clusters and the separation between them. The values of silhouette scores range from -1 to 1. A large positive score indicates that a data point is close to elements of the same cluster, a small positive score denotes a data point close to the decision boundary, while a negative score indicates that it is closer to elements in another cluster.

We generated an S-UMAP representation for our observations accounting for the stochasticity of the dimensionality reduction process with a 100-repetition distribution of silhouette scores for both the description of the individual signatures and the call types. We considered each time the combined features of the three **Bioacoustic** (20 features), **DCT** (7 features) and **MFCC** (192 features) sets (217 features in total since duration and HNR are shared by the first two sets). We computed an average silhouette score over the 100 repetitions for each call, the average score per class (or average silhouette width), the standard deviation per class, and finally the overall average silhouette score over the whole dataset (or overall average silhouette width).

As shown in Fig 5, S-UMAP representations exhibit a highly clustered pattern, both in terms of call types (left) and individual signatures (right). The degree of clustering nevertheless differs between call types and individual signatures (see below). This observation is supported by the overall average silhouette scores with, respectively, 0.94 for call types and 0.63 for individual signatures. This may be interpreted as an index of the degree of the difficulty of classifying the data points. One can thus assume intuitively that call types will be more easily discriminated than individual signatures.

Average silhouette scores per class measure the quality of each cluster. Regarding call types, peeps, with 0.9, exhibit the lowest–but still quite high–degree of clustering and barks the highest one (0.98). This validates the manual labeling of call types by human experts, and, beyond, illustrates the robustness of the adopted call type classification despite the gradient nature of the bonobo repertoire [35, 36].

Regarding individual signatures, Jill, Zuani and Zamba have silhouette scores higher than 0.9. In contrast, the scores for Lina and Djanoa's calls are close to 0. More precisely, these two animals' calls (but also Vifijo's) do not form a single cluster, but two sub-clusters, which contributes to decreasing their silhouette scores. One can further observe in Fig 5 that two soft barks produced by Bolombo, in red, overlap with Jill's cluster in brown (left upper quadrant of the graph). The interactive graph included in the file '6_Analysis_UMAP_and_Silhouette. html' available on the Github repository additionally reveals that both are extracted from the same vocal sequence (as defined in the recording procedure described in Section I). Other similar examples suggest that this pattern is not an isolated one and that when an individual exhibits extreme data points (which may or may not overlap with another cluster), these are often extracted from the same original vocal sequences.

## Automatic classification approaches and evaluation methodology

A multi-label classification task aims to assign observations described by a set of predictors to one of several predefined classes (call types or individual identities here). This task is treated here as a supervised learning task, in which a model can be trained on a set of examples with known classes, and then used to classify new ones. We chose to randomly assign 80% of the data points in our dataset to the training set, and the remaining 20% to the test set. We chose a form of discriminant analysis (DFA, see Section V for details) as a baseline, as it is a widely used classification technique in the field of animal communication, including in individual identification tasks ([17,37,66–71] among many others). We also implemented three other supervised approaches (see Section V for details), which can be described as 'state-of-the-art'

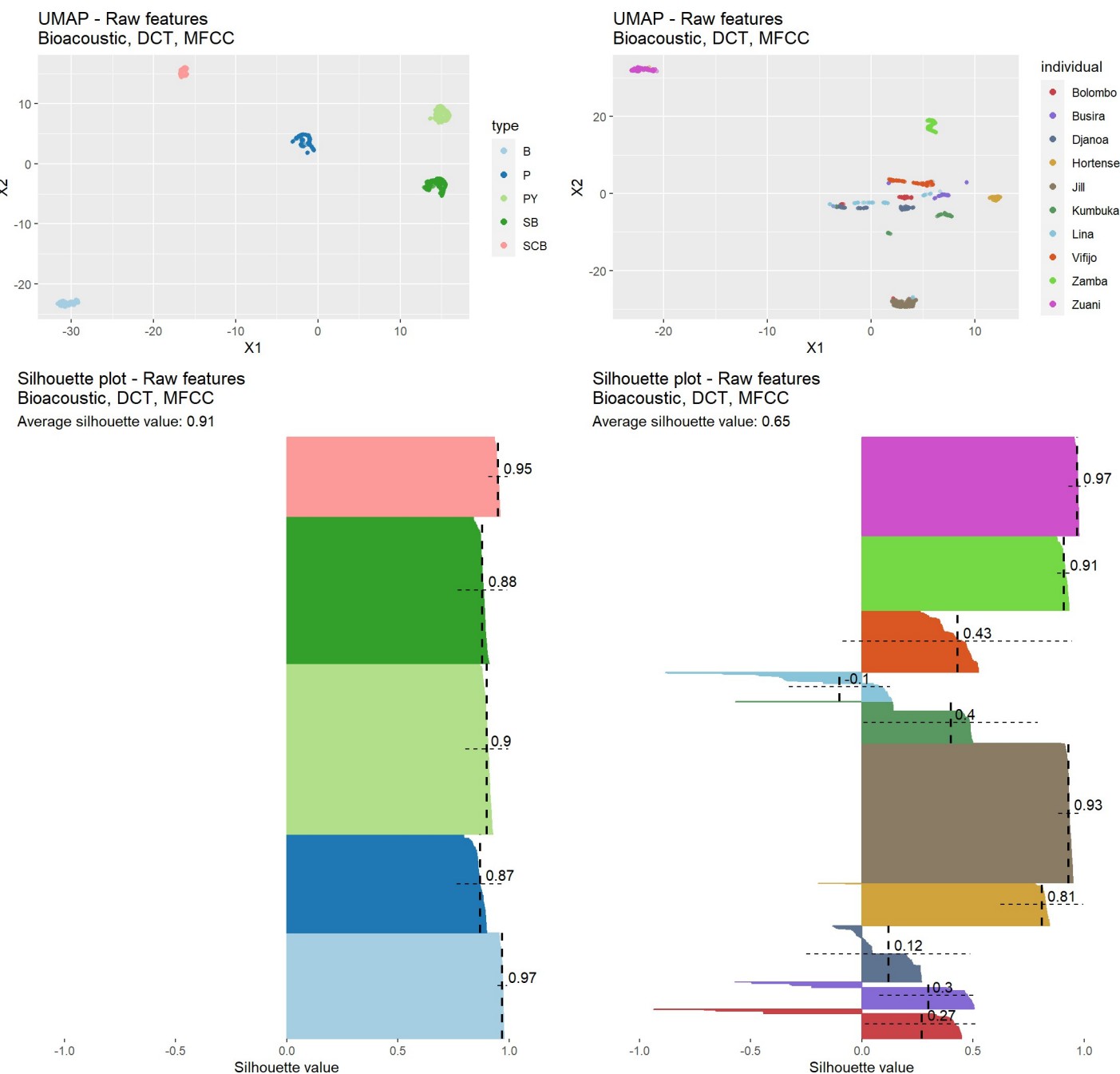

**Fig 5. Projections of bonobo calls into bidimensional acoustic spaces through S-UMAP computed on the raw acoustic features of the Bioacoustic, DCT, and MFCC sets (1,560 calls; each dot = 1 call; different colors encode different hand-labeled categories). Left. Top.** S-UMAP projection supervised by call types. **Bottom.** Silhouette profiles corresponding to the call type clustering, built from a 100-repetition distribution of silhouette scores, with averages and standard deviations per call type being represented by dashed vertical and horizontal lines, respectively. **Right. Top.** S-UMAP projection supervised by individual identities. **Bottom.** Silhouette profiles corresponding to the individual signature clustering, built from a 100-repetition distribution of silhouette scores, with averages and standard deviations per individual being represented by dashed vertical and horizontal lines, respectively.

(SOTA) in data science. SVM (Support Vector Machines) have been considered one of the best approaches for classification in the early 21<sup>st</sup> century and are widely used in classification problems, including in ecology and ethology (e.g., [72–75]). xgboost is an optimized version of

gradient tree boosting [76]—a technique which i) employs gradient descent to build a sequence of decision trees in which errors are gradually minimized and ii) boosts the influence of the best trees in the resulting forest. It is currently considered as one of the best algorithms for automatic classification or regression [77,78], even when compared to very recent deep learning methods [79] involving Neural Networks (NN). NN have been around for several decades, but their performances have improved dramatically in the last decade after the discovery of how to efficiently train deeper architectures [80]. Although they achieve by far the best performance today in computer vision and natural language processing, with models now involving up to hundreds of billions of parameters (e.g., [81]), these large networks require (very) large training datasets, which do not fit very well in the context of SUNG datasets, despite recent attempts [82]. Instead, we will consider 'shallow' dense neural networks (two to four fully connected layers, including the output layer) well suited for small datasets and the size of our different sets of predictors, as they have proven effective in similar applications (e.g., [3,83]).

Each of our SOTA' approaches involves the tuning of a number of hyperparameters (see Section V for details), which values can impact performance on a given dataset [84]. While the parameters (e.g., the values of the connections between neurons in a NN) are adjusted during the training phase, the hyperparameters (e.g. the number of layers and the number of neurons in each layer of the NN) are not and have to be specified otherwise. We have implemented the usual approach in machine learning, by taking off a part of the training set (the so-called validation set) to find the optimal values of the hyperparameters for the data at hand, in a process known as hyperparameter tuning.

To evaluate the performance of our different classification techniques, we had to specifically consider the imbalance in the dataset (see Section V). For our SOTA approaches, we first did so by assigning, to each category, a weight inversely proportional to its number of members, as a way to counterbalance the under-represented data. For DFA we followed previous studies to build reduced balanced training sets instead (see Section V). Second, we considered a metric adapted to an imbalanced dataset, since not all metrics are appropriate in this situation. In particular, the standard accuracy, which is easily interpreted, returns results biased towards the more represented classes. We therefore considered three measures in addition to **standard accuracy** (which we kept for the sake of comparison with previous studies): multi-class log loss (a.k.a. cross-entropy), multi-class Area Under the Receiver Operating Characteristics Curve (known as AUC) and balanced accuracy:

– Multi-class **log loss** penalizes the divergence between actual and predicted probabilities—lower values are better. Log loss differs from the next two metrics in that it considers probabilities rather than classification outputs.

– Multi-class **AUC** [85] extends (two-class) AUC with two possible binarization strategies of the multiple-class problem: i) reducing it to several one-versus-all-others problems or ii) reducing it to several one-versus-one problems (results can be averaged in both cases). We adopted the second option while considering additionally the a priori distribution of the classes to better address imbalance.

– The **balanced accuracy** (bac) is defined as the average of recall (a.k.a. Sensitivity) obtained on each class. This addresses the issue of standard accuracy being biased by classes of different sizes. Balanced accuracy offers the advantage of being straightforward to understand, compared to log loss and AUC.

A pair of training set and test set leads to a single value for each of these four indices. However, such estimators of a classifier's performance are sensitive to the exact composition of

each set and can thus under- or overestimate the true (unobservable) performances. To minimize this issue, we implemented a standard procedure consisting in repeating 100 times the whole process of random training-test set creation to introduce fluctuations in the set compositions. For each performance index, it resulted in a distribution of 100 values whose mean and standard deviation provided robust estimators of the classifier performance.

We further evaluated the importance (averaged over the 100 iterations) of the different features used as predictors when classifying the calls, in order to detect whether some of them play a significantly larger role than others. We analyzed the features of the feature sets 'Bioacoustic' and 'DCT', but did not consider MFCC as their large number leads to very limited– and hardly meaningful–impacts for the individual features (see Section V for details).

In addition to the previous iterated approach, to assess whether performances were significantly above chance, we computed a random baseline based on a Monte Carlo permutation test [86]. It consists of 1,000 instances of permutation resampling, i.e.,1,000 pairs of randomly drawn training-test sets, with each time a prior nested random shuffling of the values of the predicted variable (PV)–call type or emitter identity, depending on the task. Nested random sampling means here that the random shuffling of the PV occurs within each level of the 'secondary' variable, that is emitter identity when the PV is call type, and call type when the PV is emitter identity. Taking into account the interaction between these two variables and the unbalanced distribution of the data, this sampling leads to a conservative estimate of the baseline [41,87]. For each performance measure, the 1,000 pairs of training and test sets lead to a distribution of performance values under the null hypothesis (i.e., no relationship between the predictors and the predicted variable). The empirical $p$-value of the performance achieved with the original analysis can then be obtained by dividing the number of randomized datasets which lead to an equal or better performance than the non-permuted configuration by the total number of datasets tested, i.e., 1,000. The average performance for the 1,000 iteration further provides a robust estimate of the chance performance [17,37].

Different feature sets and classifiers may differently model the information present in the dataset. This suggests that they can be combined to achieve a better performance by cumulating their individual strengths while mitigating their individual weaknesses. So-called *ensembling* methods have been successfully developed for a large number of machine learning challenges. Three popular methods to combine different models are i) bagging—with models built with different subsamples of the training dataset -, ii) boosting—with models chained to gradually reduce prediction errors (like in xgboost) -, and iii) stacking—with several parallel models applied to the same observations [88]. Different stacking approaches are available: simple ones like voting or averaging predictions, and more advanced ones which involve additional supervisor models—known as super learners—using the predictions of the initial models as inputs. We implemented stacking with a penalized linear regression as super learner, in order to account for the strong correlation existing between the outputs of the base models (see Section V for details).

Once an ensemble learner is defined, its performances can be evaluated exactly in the same way as non-ensemble ones, allowing one to estimate the gain of the ensembling strategy. We defined and implemented a) three stacked learners combining all feature sets for each of the three classifiers, b) three stacked learners conversely combining the three classifiers for each feature set, and finally c) a seventh configuration stacking all combinations of feature set and classifier (full description can be found in 'Ensembling' in Section V) and tested them along with the individual classifiers.

We mainly report results with balanced accuracy below, as it is more directly interpretable than log loss and AUC. More details about the approach can be found in Section V, as well as in the files of the Github repository. In particular, the full code to replicate the analysis is

provided. Importantly, a streamlined version of the proposed workflow is also provided as a distinct file in order to propose a procedure easily adaptable to another dataset.

## III. Automatic classification: Results

### Task 1: Identification of call types

This task consists in classifying each call from the test set as belonging to one of the five categories listed in Table 1, regardless of the individual who produced it. Our results (Fig 6 and Table 2) confirm that the five call types considered are discriminable to some extent, with a balanced accuracy reaching 0.794 with the best classifier. In comparison, the chance level is equal to 0.200.

The three classifiers SVM, NN, and xgboost outperform the DFA approach, both with the bioacoustic set and the DCT set, and independently of the metric considered. DFA therefore partially misses some of the discriminative information available in the acoustic features. The balanced accuracy obtained with DFA is indeed only 0.596 with the bioacoustic set. This performance is comparable to that obtained by [37] with the same method (57% of accuracy in a 5-category task with a slightly different call type labeling).

To compare the results of the discriminant analyses to the chance level, modified datasets were created by recombination (see Section V) and DFA applied to them. This 1,000-fold iterated procedure provided a robust estimate of the distribution of random accuracies. The empirical p-values obtained after this recombination procedure were equal to p = 0.001.

Leaving aside stacked learners, it can also be seen that i) the results obtained with the MFCC set are worse than those obtained with the bioacoustic or DCT sets, ii) for a given feature set, xgboost and svm reach very comparable performances, better than those obtained with the NN approach. The best performing configuration combines xgboost with the bioacoustic set though, as it tends to outperform the combination of svm with the bioacoustic set (0.747 vs 0.718 respectively). Although the MFCC set carries a richer description of the calls, using it does not bring any advantage, and even degrades the performance of the classifiers. The dimensionality of our MFCC representation is quite large (192 dimensions) and rather atypical. A thorough discussion is beyond the scope of this paper, but a more traditional representation based on performing a cepstral analysis at the middle of the call and adding delta and delta-delta coefficients performs worse (See S1 Text for more details). The better performance achieved with the bioacoustic set is consistent with the fact that the bioacoustic features are the cornerstone on which each call type is primarily defined by expert bioacousticians and primatologists (e.g., [87,89,90] for recent perspectives). Finally, the fact that the performance reached with the DCT set is almost as good as with the bioacoustic set is very encouraging: it indicates that a small number of acoustic descriptors succeed in capturing most of the relevant information present in the signal.

When it comes to ensembling, all seven configurations improve the performance of the classifiers (or learners) they build upon. The best results are obtained with the stacking of all nine learners. The improvement is once again obvious, especially compared to the DFA approach.

Furthermore, the observed difference between accuracy and balanced accuracy tends to be smaller for the stacked classifiers than for each algorithm separately, suggesting that the former handles class imbalance better. Comparison to the random baseline showed that all results (for all metrics considered) are significantly above chance level with $p < 0.001$.

Focusing on the best performing approach–the stacking of the 9 different configurations– Fig 7 displays the average confusion matrix for 100 iterations. It confirms the quality of the classification, but also highlights that the risk of confusion is the highest for B, which is

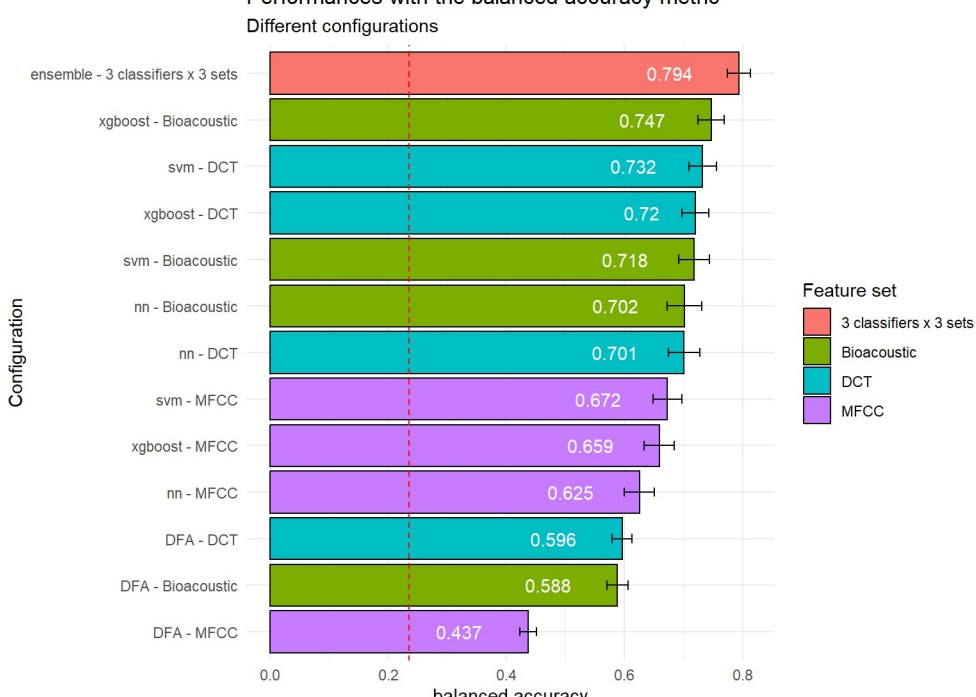

**Fig 6. Performance in classifying bonobo call types as a function of classifier and acoustic set used.** The red bar shows the performance achieved by an ensemble classifier combining the 9 primary classifiers. The other bars correspond to configurations associating each classifier with different sets of acoustic features (Bioacoustic, DCT, MFCC). The configurations are sorted by decreasing performance from top to bottom. Performance is reported in terms of balanced accuracy. Green, turquoise, and purple indicate the models trained on the Bioacoustic, DCT, and MFCC feature sets respectively. Chance level is represented by the vertical dashed red line. The error bars report the standard deviation of the performances for the 100 iterations of the evaluation process.

relatively often confused with SB, SCB and PY, while P and PY are the most easily identifiable calls. Confusion thus occurs mainly between call types that are "adjacent" in terms of duration. Additionally, one can note that the implemented methodology seems quite robust to class imbalance, the two worst performances (72.2% and 73.3% of correct classification) being reached with two of the most frequent call types B and SB respectively).

Fig 8 shows the relative importance of the different acoustic descriptors as estimated with xgboost. Duration appears to be the most important feature, followed by f0.onset and f0.offset. For the DCT approach, dct2 –related to the curvature of the $f_0$ trajectory–and duration are the two major predictors. A detailed comparative analysis of the two feature sets would fall beyond the scope of this paper because it would be excessively specific to the bonobo case, but one can mention two interesting aspects. First, the importance of the duration is similar in the two sets. Secondly, a global shape factor such as dct2 in the DCT set seems to capture an information (the curvature) that is spread over several dimensions in the Bioacoustic set. It confirms that the call shape itself is salient and relevant, meaning that the covariation of the bioacoustic features may be more important than the small acoustic fluctuation that can locally affect each feature. The SVM and NN approaches give similar information. These results show that call types can be characterized by very few acoustic descriptors.

**Table 2. Metrics characterizing the classification performance of call types as a function of the classifier and acoustic set used.** Four metrics are reported: log loss, AUC, balanced accuracy, and accuracy. The best performance achieved by a primary configuration (upper part) and an ensemble configuration (lower part) is displayed in bold. For AUC, accuracy (acc) and balanced accuracy (bac), a color scale highlights the progression from the lowest scores (in pale orange) to the highest scores (in dark orange) in the column.

| Algorithm | Feature set | Config. # | log loss | AUC | acc | bac |
|---|---|---|---|---|---|---|
| DFA | Bioacoustic | | 0.819 | 0.894 | 0.651 | 0.588 |
| DFA | DCT | | 0.813 | 0.901 | 0.674 | 0.596 |
| DFA | MFCC | | 1.310 | 0.779 | 0.491 | 0.437 |
| svm | Bioacoustic | (1) | 0.671 | 0.931 | 0.736 | 0.718 |
| svm | DCT | (2) | 0.658 | 0.934 | 0.745 | 0.732 |
| svm | MFCC | (3) | 0.810 | 0.906 | 0.664 | 0.672 |
| nn | Bioacoustic | (4) | 0.674 | 0.933 | 0.728 | 0.702 |
| nn | DCT | (5) | 0.681 | 0.932 | 0.724 | 0.701 |
| nn | MFCC | (6) | 0.890 | 0.888 | 0.621 | 0.625 |
| xgboost | Bioacoustic | (7) | **0.627** | **0.940** | **0.755** | **0.747** |
| xgboost | DCT | (8) | 0.651 | 0.936 | 0.732 | 0.720 |
| xgboost | MFCC | (9) | 0.830 | 0.901 | 0.654 | 0.659 |
| ensemble | 3 sets, svm | (1+2+3) | 0.581 | 0.952 | 0.785 | 0.784 |
| ensemble | 3 sets, nn | (4+5+6) | 0.604 | 0.950 | 0.773 | 0.767 |
| ensemble | 3 sets, xgboost | (7+8+9) | 0.578 | 0.952 | 0.785 | 0.784 |
| ensemble | 3 classifiers, Bioacoustic set | (1+4+7) | 0.640 | 0.939 | 0.759 | 0.744 |
| ensemble | 3 classifiers, DCT set | (2+5+8) | 0.648 | 0.938 | 0.752 | 0.736 |
| ensemble | 3 classifiers, MFCC set | (3+6+9) | 0.792 | 0.912 | 0.675 | 0.683 |
| ensemble | 3 classifiers x 3 sets | (1+2+3+4+5+6+7+8+9) | **0.542** | **0.957** | **0.794** | **0.794** |

## Task 2: Identification of Individual signatures

This task consists in assigning each call from the test set to the individual who produced it out of the ten bonobos listed in Table 1, regardless of the call category. The chance level corresponds to a bac equal to 0.100. The best performance (bac = 0.507) in this 10-class problem is lower than for the 5-class call type classification. However, it is again much better than that given using DFA (bac = 0.236). The difference in performance between the DFA approach and the other approaches is even greater when it comes to identifying individuals than for call types (Fig 9 and Table 3). The three classifiers SVM, NN and xgboost again outperform DFA with both bioacoustic and DCT sets, regardless of the metric considered.

Leaving stacked learners aside, the best performance is obtained with the MFCC set, then the bioacoustic set and finally the DCT. For the three feature sets, the best performing classifier approach is consistently xgboost, but the same level of performance is achieved with svm when MFCC are considered. Contrary to what we found with the call types, the richness of the MFCC description enhances discrimination between the individual signatures. This result suggests that the bonobo vocal signature results from salient differences in the way each individual arranges its calls (as illustrated in Fig 4 by the differences observed between the templates of individuals #19 and #20), complemented by subtle variations more easily captured by MFCC than by standard bioacoustic features. Conversely, DCT representation only approximates the tonal contour augmented with harmonicity of the call. It thus probably fails to account for such subtle differences in individual voices, even if it still captures individual differences in the relative position of the calls.

When it comes to ensembling, six of the seven configurations improve the performance of the learners on which they are based while stacking the three algorithms NN, SVM and

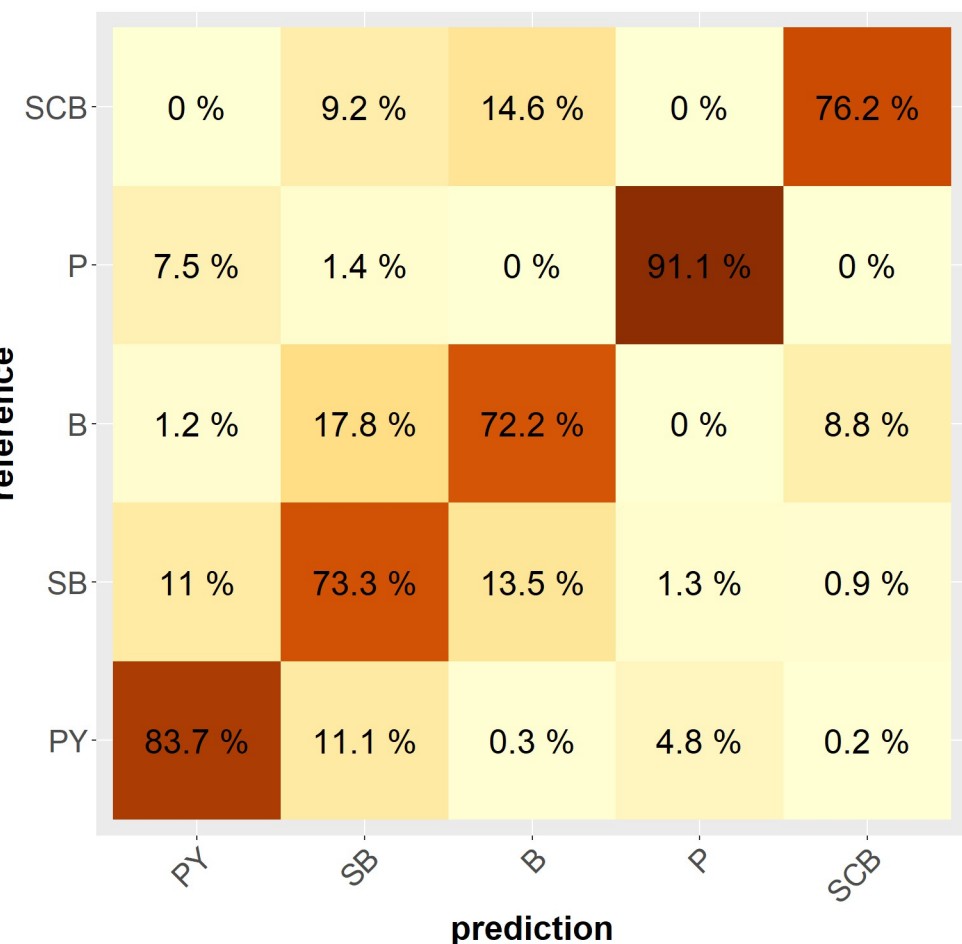

**Fig 7. Average confusion matrix, for 100 iterations of the evaluation process, reporting the classification rates of the call types in the best configuration (the ensemble classifier combining the 9 primary classifiers).** Types are sorted from bottom to top by decreasing number of occurrences (PY: most frequent; SCB: least frequent). Percentages are according to the reference and sum to 1 along rows. The value of a cell color is proportional to its percentage (the darker, the larger).

xgboost with the same set of bioacoustic features does not bring any improvement. This suggests a ceiling effect. The best results are obtained with the stacking of all 9 configurations.

As with call types, all classification results (whatever the metrics considered) are significantly above chance level with $p < 0.001$. However, the impact of the unbalanced dataset is striking. With the ensemble configuration leading to the best performance–the stacking of the nine different configurations–we obtain, on the one hand, quite good performances (up to 79.8% correct identification) for the four individuals contributing the most to the dataset (Jill, Zuani, Zamba, Vifijo) ([Fig 10]). On the other hand, the performances are modest, though above chance, for the individuals that contribute less (e.g., Bolombo = 17.3% of correct identification; Busira = 28.1%). Class imbalance thus has a significant impact on our results, despite the adoption of class weights to mitigate it. These results suggest that when a poor individual classification score is obtained, it is likely to be due to a faulty classifier and not to the absence of idiosyncratic features in an individual's calls.

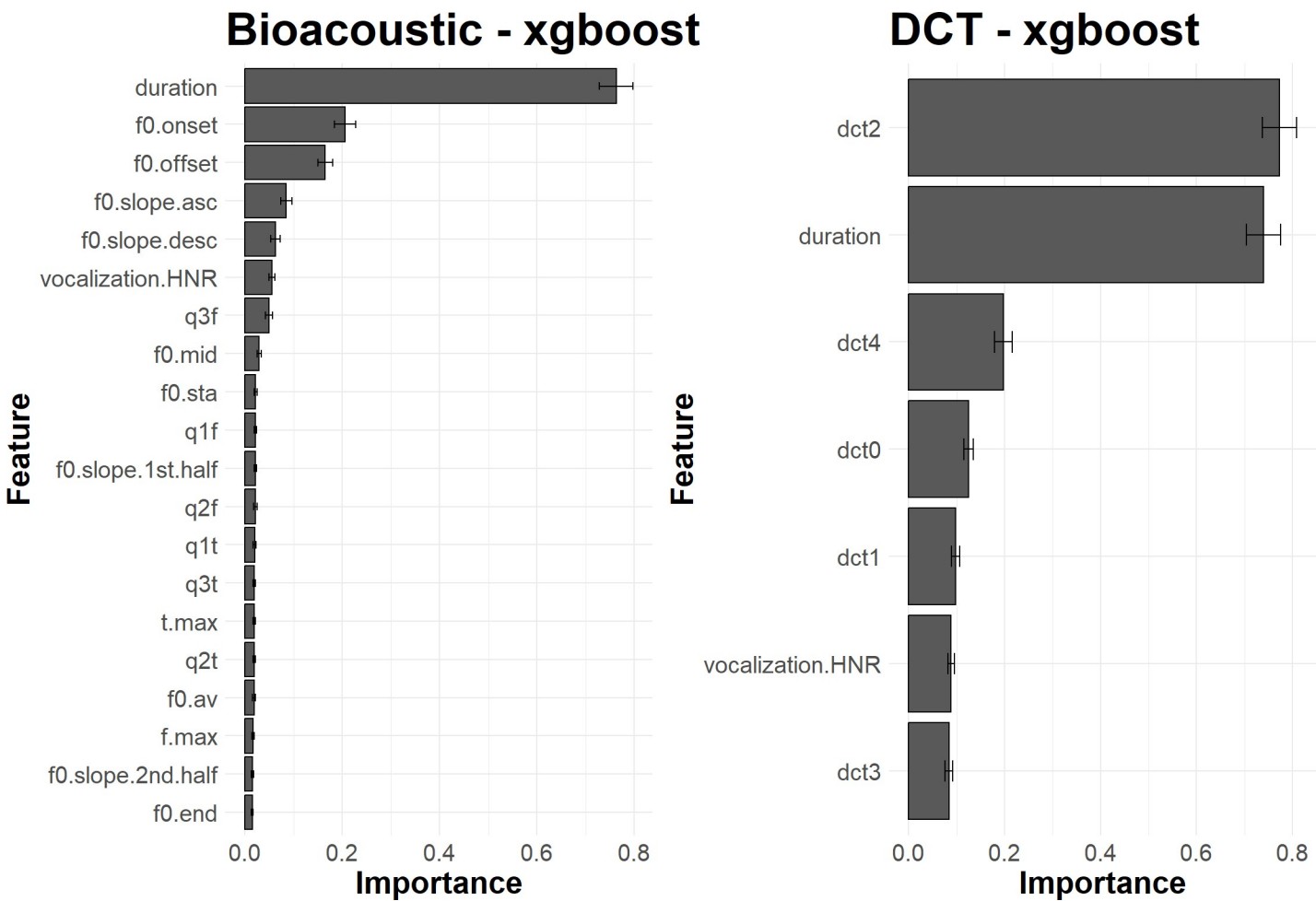

**Fig 8. Average importance of acoustic features, for 100 iterations of the evaluation process, when classifying call types with xgboost. Left**. Features of the Bioacoustic set. **Right**. Features of the DCT set. The bar plots illustrate the relative influence of each acoustic feature on the classification performance. The error bars report the standard deviation of the measure of importance for the 100 iterations of the evaluation process.

By examining the impact of each feature on classification performance (Fig 11), it can finally be observed that their importance is more diffuse across a wider set of features than was observed in the call type classification task.

## IV. Addressing possible data leakage

Data leakage refers to the situation where a classification decision is based on information made coincidentally available to the classifier during its training phase. Let's imagine two primates A and B vocalizing in an outdoor enclosure with different vocal behaviors regarding the observer. A is "shy" and stays at a distance from the audio recorder while B, being much more curious, is eager to come close to the recorder. For an equivalent vocal effort by the two individuals, intensity levels will differ between recordings from A and B. In absence of intensity normalization in postprocessing, an automatic classifier can easily pick this information available in the training set and correctly discriminate between A and B based on their distance to the recorder and not on their acoustic signature. The classifier decision is thus contaminated by a confounder of the true task of interest–the position of the emitters rather than their

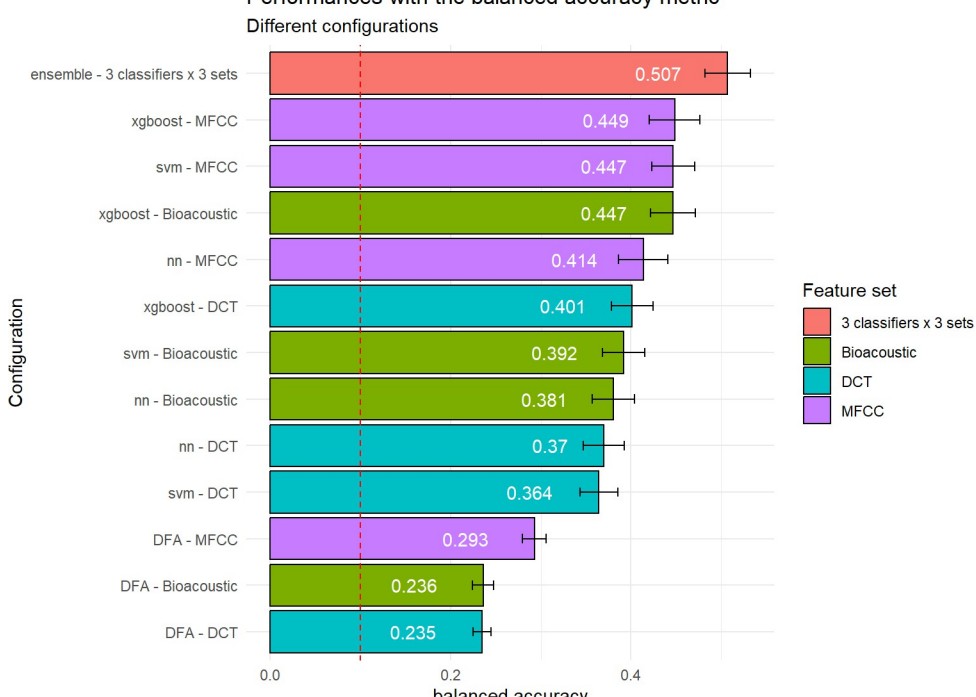

**Fig 9. Performance in classifying bonobo individual signatures as a function of classifier and acoustic set used.**
The red bar shows the performance achieved by an ensemble classifier combining the 9 primary classifiers. The other bars correspond to configurations associating each classifier with different sets of acoustic features (Bioacoustic, DCT, MFCC). The configurations are sorted by decreasing performance from top to bottom. Performance is reported in terms of balanced accuracy. Green, turquoise, and purple indicate the models trained on the Bioacoustic, DCT, and MFCC feature sets respectively. Chance level is represented by the vertical dashed red line. The error bars report the standard deviation of the performances for the 100 iterations of the evaluation process.

intrinsic characteristics. In this simplified example, the issue is easily detected, but real situations can involve much subtler forms of leakage that classifiers can exploit with undue success.

When a classification evaluation is performed on large datasets, a careful experimental design can prevent from falling in the most obvious traps, but other sources of leakage can still prove problematic (for more information, please refer to [13] for a general discussion, [47] for a methodological approach in animal species acoustic monitoring, and [91] for an evaluation in music classification). The situation is compounded in the context of SUNG datasets: by definition, they don't offer a faithful sample of all possible situations of vocalization but a degraded, albeit informative, perspective. In this section we illustrate two dimensions examined in our case study. In the first subsection, we show how a leakage issue related to the occurrence of calls in sequences can be identified and mitigated in the context of vocal signature recognition. In the second subsection, we examine how the data structure leads to the potential presence of confounders that cannot be efficiently ruled out with a SUNG dataset.

## The vocal sequence effect

In the automatic classification approach reported in the previous section, each observation unit consists of a single call. A problem may arise when these calls are extracted from the same vocal sequence (as defined in the recording procedure described in Section I), which is frequently the case in animal datasets. This situation violates the independence of the observations (pseudoreplication) and can potentially undermine the validity of the classification

**Table 3. Metrics characterizing the classification performance of individual signatures as a function of the classifier and acoustic set used.** Four metrics are reported: log loss, AUC, balanced accuracy, and accuracy. The best performance achieved by a primary configuration (upper part) and an ensemble configuration (lower part) is displayed in bold. For AUC, accuracy (acc) and balanced accuracy (bac), a color scale highlights the progression from the lowest scores (in pale orange) to the highest scores (in dark orange) in the column.

| Algorithm | Feature set | Config. # | log loss | AUC | acc | bac |
|---|---|---|---|---|---|---|
| DFA | Bioacoustic | | 1.745 | 0.731 | 0.410 | 0.236 |
| DFA | DCT | | 1.698 | 0.731 | 0.429 | 0.235 |
| DFA | MFCC | | 1.785 | 0.732 | 0.457 | 0.293 |
| svm | Bioacoustic | (1) | 1.490 | 0.840 | 0.516 | 0.392 |
| svm | DCT | (2) | 1.567 | 0.822 | 0.486 | 0.364 |
| svm | MFCC | (3) | 1.416 | 0.855 | 0.548 | 0.447 |
| nn | Bioacoustic | (4) | 1.487 | 0.839 | 0.508 | 0.381 |
| nn | DCT | (5) | 1.537 | 0.829 | 0.490 | 0.370 |
| nn | MFCC | (6) | 1.415 | 0.855 | 0.536 | 0.414 |
| xgboost | Bioacoustic | (7) | 1.415 | 0.850 | 0.544 | 0.447 |
| xgboost | DCT | (8) | 1.530 | 0.827 | 0.493 | 0.401 |
| xgboost | MFCC | (9) | 1.366 | **0.861** | **0.552** | **0.449** |
| ensemble | 3 sets, svm | (1+2+3) | 1.240 | 0.887 | 0.594 | 0.495 |
| ensemble | 3 sets, nn | (4+5+6) | 1.264 | 0.886 | 0.583 | 0.466 |
| ensemble | 3 sets, xgboost | (7+8+9) | 1.254 | 0.885 | 0.589 | 0.484 |
| ensemble | 3 classifiers, Bioacoustic set | (1+4+7) | 1.420 | 0.851 | 0.542 | 0.427 |
| ensemble | 3 classifiers, DCT set | (2+5+8) | 1.493 | 0.835 | 0.500 | 0.389 |
| ensemble | 3 classifiers, MFCC set | (3+6+9) | 1.311 | 0.873 | 0.571 | 0.465 |
| ensemble | 3 classifiers x 3 sets | (1+2+3+4+5+6+7+8+9) | 1.210 | **0.894** | **0.605** | **0.507** |

performance. Specifically, how can we be sure that certain features characterizing the call sequence as a whole are not used by the classifier to identify the emitter? If this is the case, as suggested for instance by [7], single call classification performance could be overestimated.

To address this issue, we compared three different subsampling scenarios to build training and test subsets. The first one (called *Default*) corresponds to the results reported in Section II. It simply consists in not exercising any control over how calls are assigned to one or the other subset, other than to ensure similar distributions of the occurrences of individuals in both sets. A second one (*Fair* scenario) consists in minimizing overlap (i.e., calls belonging to the same vocal sequence) by assigning as many sequences as possible to either the training set or the test set, so that the soundscapes of the sequences seen during training do not provide any information when classifying calls in the test phase. This optimization was performed with an in-house tool called BaLi (see 'Evaluating Data leakage' in Section V for details). Full independence can be achieved in theory if enough data sequences are available to match the distributions of types and individuals between the training and test sets, but in practice the limited size of the dataset leads to residual leakage (see results below).

Finally, the third scenario (*Skewed*) consists in maximizing the proportion of sequences shared by both sets (but still with disjoint call sets). By definition, the *Skewed* scenario is ill-designed as it maximizes data leakage, which automatically leads to an overestimation of classification performances. It is nevertheless instructive in providing ceiling performance against which the *Default* and *Fair* scenario can be compared.

For the sake of simplicity, we only consider the classification of individual signatures, and not the one of call types.

To assess the influence of data leakage, we followed the resampling strategy described in 'Automatic classification approaches and evaluation methodology' in Section II and drew the

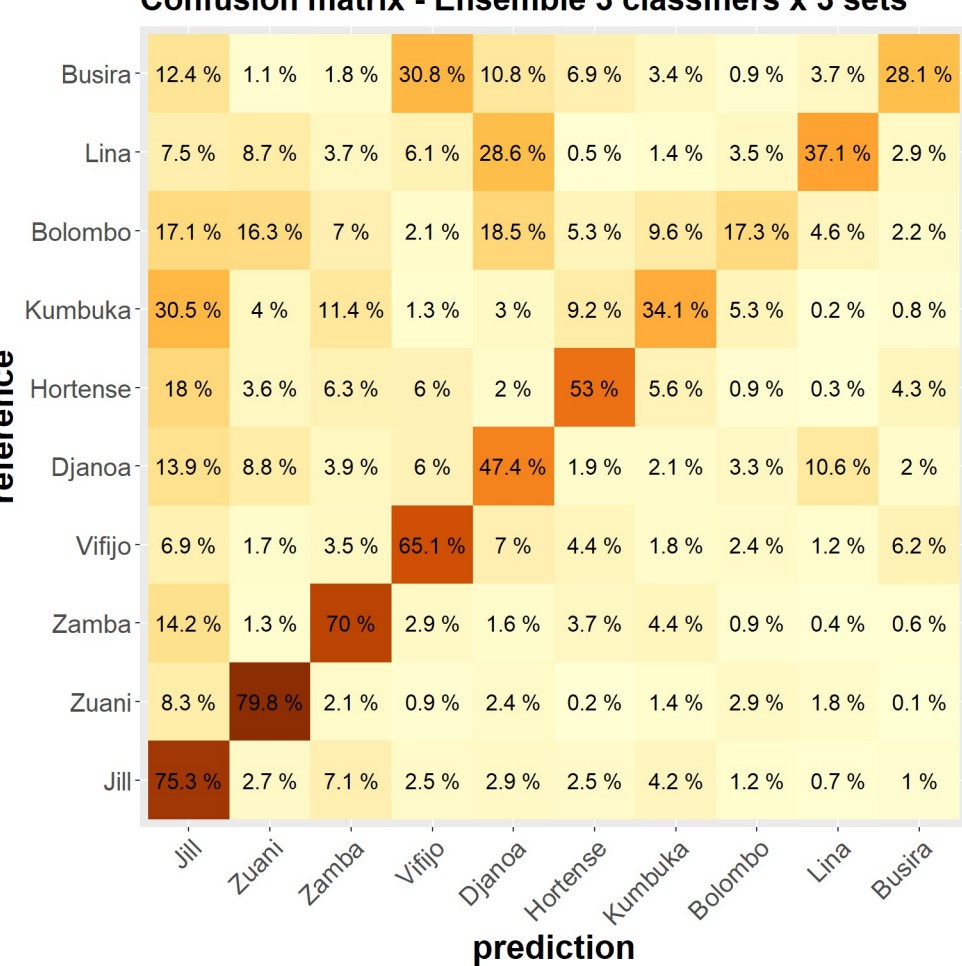

**Fig 10. Average confusion matrix, for 100 iterations of the evaluation process, reporting the classification rates of the individual signatures in the best configuration (the ensemble classifier combining the 9 primary classifiers).** Individuals are sorted from bottom to top by decreasing the number of calls (Jill: largest number; Busira: lowest number). Percentages are according to the reference and sum to 1 along rows. The value of a cell color is proportional to its percentage (the darker, the larger).

training and test sets 100 times following each sampling scenario. The outputs of our approach are displayed in Fig 12. On the left side, the horizontal axis corresponds to a measure of the degree of overlap, defined as the number of call swaps required for all sequences to appear in a single set (ignoring the constraint that the training set should be four times larger than the test set because of the 80%-20% split of the whole dataset). The count value is thus equal to zero for an ideally fair split without overlapping sequences between the training and test sets. It can be seen that doing nothing (Default) is actually closer to maximizing overlapping (Skewed) than to minimizing it (Fair).

We hypothesize that the performance would be the highest for the Skewed scenario and the lowest for the Fair one. In addition to the 100 runs reported in the previous section following the Default sampling, we computed 100 runs for both the Fair and Biased scenarios. For the sake of simplicity, we left aside ensemble learners and focused on our 9 initial configurations.

The results can be found on the right side of Fig 12. Our hypotheses are confirmed, i.e., preventing overlapping of the sequences leads to reduced performances, when maximizing it

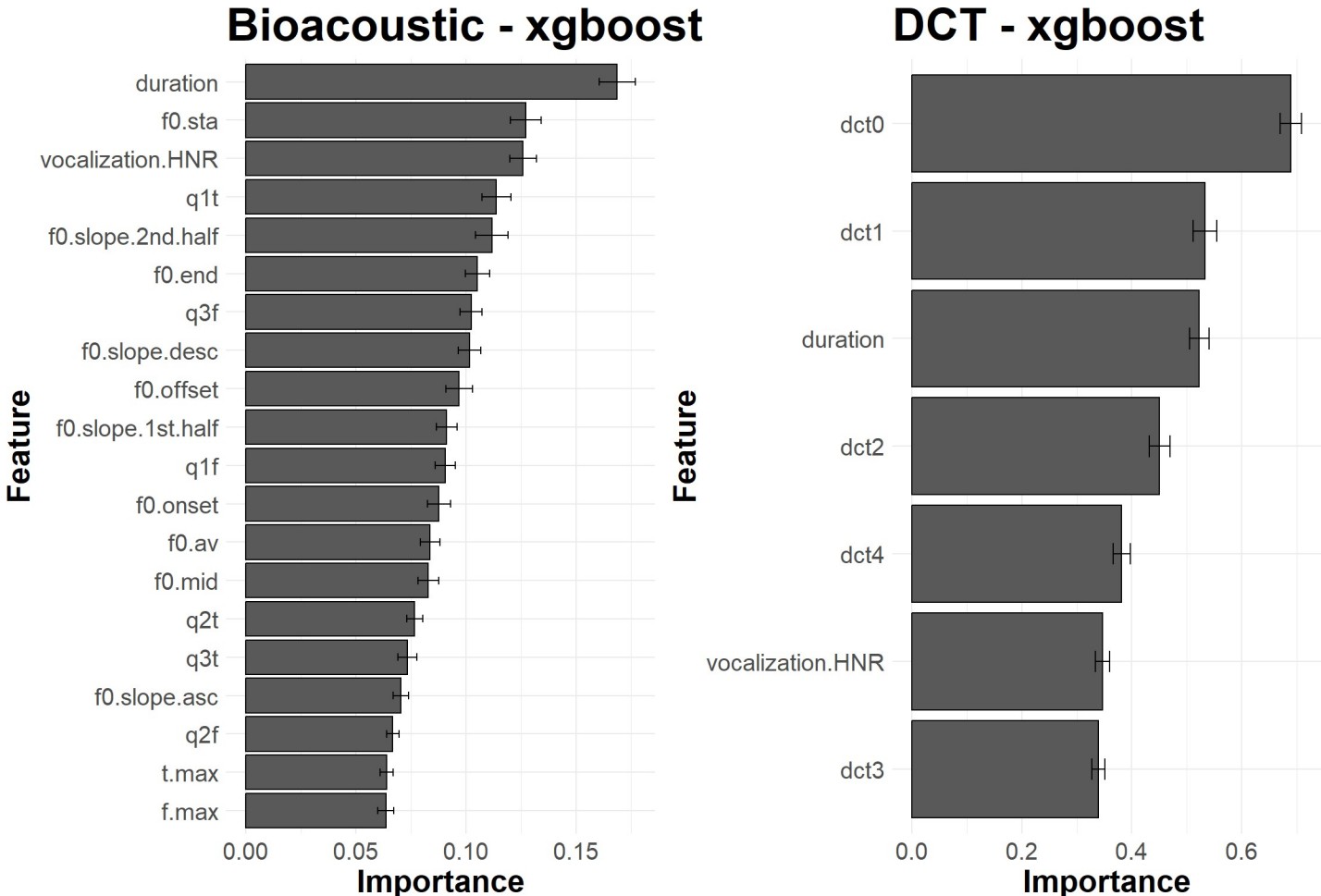

**Fig 11. Average importance of acoustic features, for 100 iterations of the evaluation process, when classifying individual signatures with xgboost. Left**. Features of the Bioacoustic set. **Right**. Features of the DCT set. The bar plots illustrate the relative influence of each acoustic feature on the classification performance. The error bars report the standard deviation of the measure of importance for the 100 iterations of the evaluation process.

leads to inflated ones. The former are more reliable as they correspond to minimization of the issue of non-independence between the observations where it matters most, i.e., between the training set and the testing set. One can observe, however, that the differences between the different strategies are small, which raises the question whether it is really necessary to control for the grouping of calls in sequences. Additionally, there are no differences in the general pattern of performances across classifiers and sets of predictors. Results are similar across our different metrics (see the file '5_Analysis_Classification_Performances.html' in the Github repository).

One explanation to the previous observations may lie in how calls are specifically organized in sequences in our dataset. Out of the 571 sequences in our dataset of 1,560 calls, 259 sequences consist of only one call, 111 consist of 2 calls and 201 of more than 2 calls. The calls in a sequence can be of the same or of different types. This may explain why the differences between the different scenarios are limited: by definition, a one-call long sequence cannot be shared by the training and test sets.

To further test this possibility, we built a subset consisting only of calls appearing in sequences of at least 3 elements–all 10 individuals were still present for a total of 1,079 calls in 201 sequences. We then followed the same approach as with our primary dataset, considering

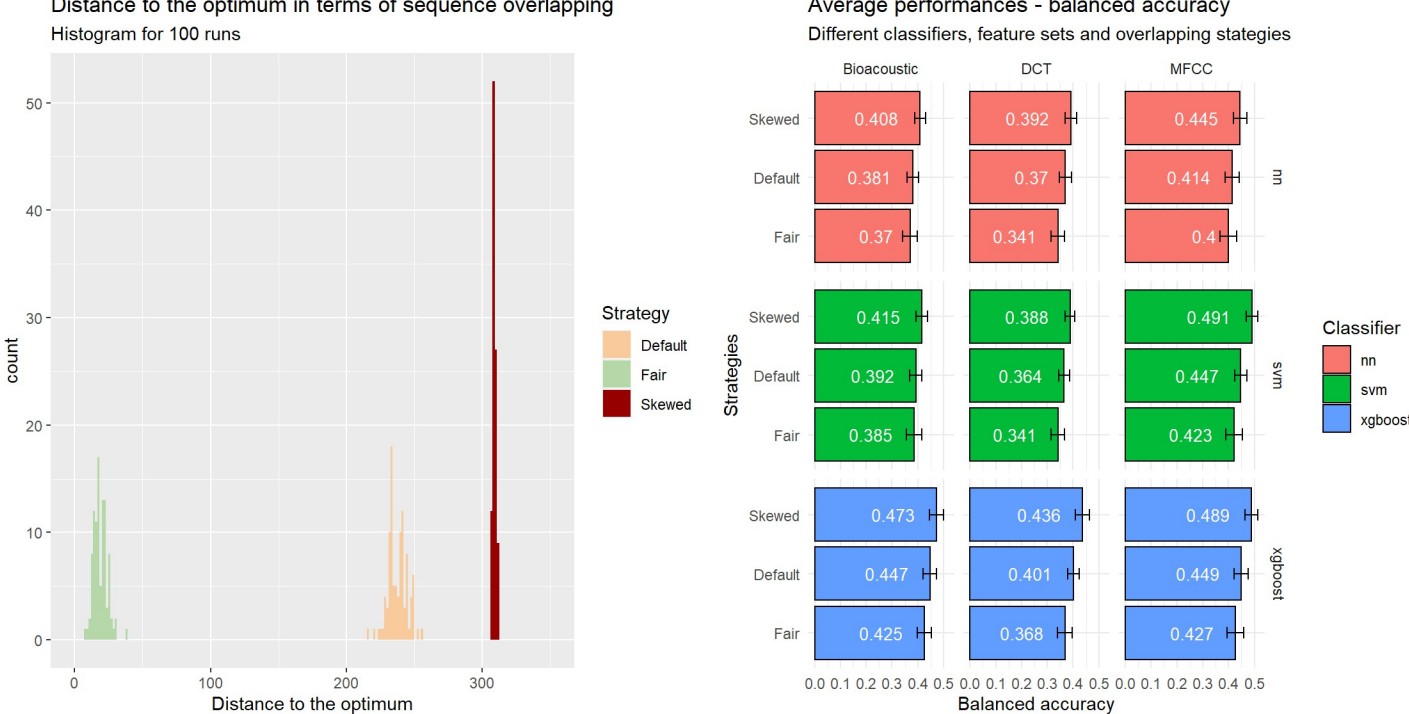

**Fig 12. Influence of the sampling on data leakage (all sequences considered).** Three scenarios are applied: *Default*, *Fair* and *Skewed*. **Left.** Distribution of the 100 runs for each strategy in terms of sequence overlap between training and test sets (0: no overlap). **Right.** Influence of scenario on performance (balanced accuracy) for each combination of classifiers and acoustic feature sets when classifying individual signatures.

our three different scenarios with 100 runs for each, after having estimated the best hyperparameters for our 9 different configurations of sets of predictors and classifiers. Fig 13 reports the results as in Fig 12. While the overall pattern is unchanged, one can notice on the left side of the figure that the Default scenario is very close to the Skewed one, meaning that the 'simplest' approach is close to the worst scenario in data leakage.

In terms of classification performance (Fig 13, right), one can observe larger differences between the Fair scenario and the others (e.g., a 12.5% gap in balanced accuracy between Fair and Default for xgboost with MFCC). It highlights that performance is clearly overestimated when the classifier can extract information that would be inaccessible in real life conditions.

The conclusion of our investigations is a cautionary tale: the occurrence of calls in sequences should be controlled to prevent the overestimation of classification performances, and this all the more as the average number of calls per sequence increases. The Skewed scenario should definitely be avoided since it dramatically overestimates the classification performances. On the contrary, one should try to implement a Fair strategy in order to get the least-biased performance estimation. If done manually, it can be tenuous, but optimization tools such as Bali provide an efficient way to alleviate the combinatorial burden (see 'Evaluating data leakage / Bali' in Section V). The Default strategy merely consists in the absence of any strategy and it is arguably a common practice when data leakage is–wrongly or rightly–not identified as a problem. One should mention that when data paucity is not an issue, practitioners often adopt a conservative approach that avoids most data leakage, for instance by systematically assigning recordings done on different dates to the training and test sets. Our caveat is thus mostly relevant in the context of SUNG datasets.

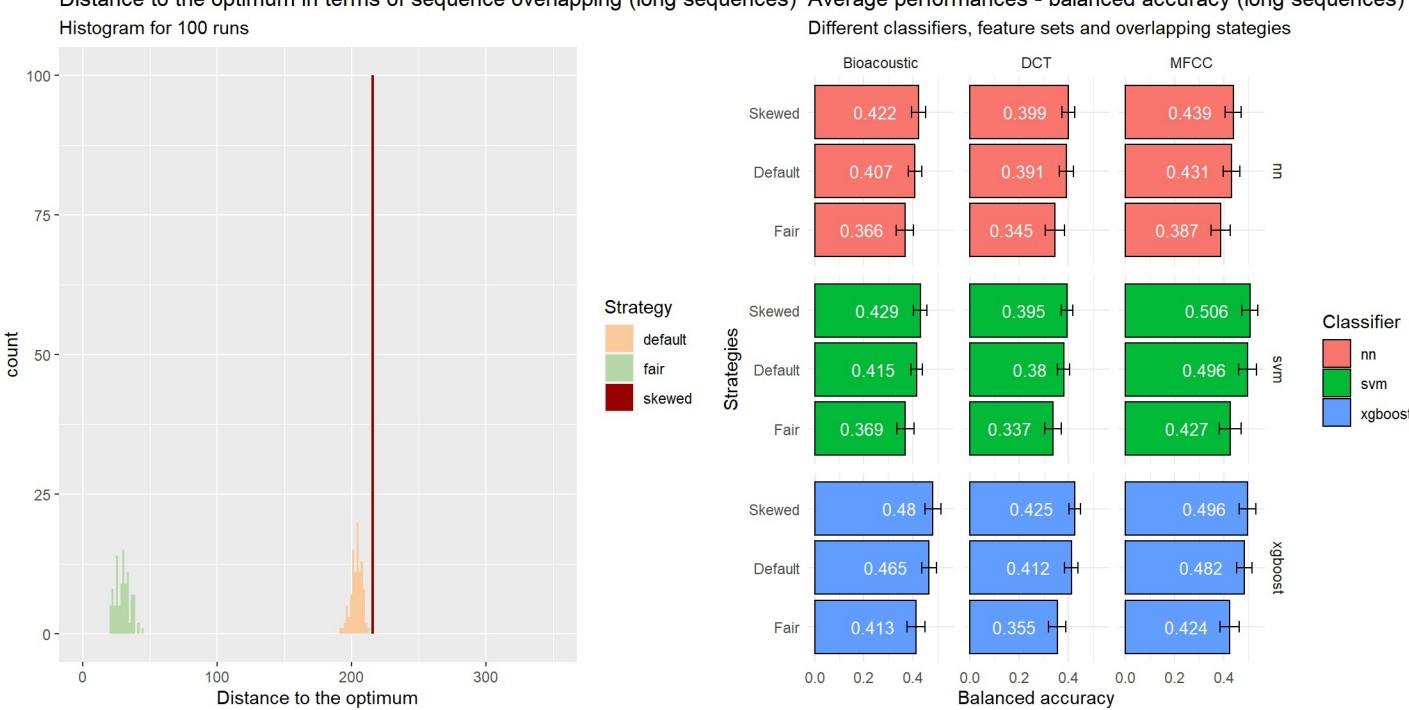

**Fig 13. Influence of the sampling on data leakage (sequences with at least three calls considered).** Three scenarios are applied: *Default*, *Fair* and *Skewed*. **Left.** Distribution of the 100 runs for each strategy in terms of sequence overlap between training and test sets (0: no overlap). **Right.** Influence of strategy on performance (balanced accuracy) for each combination of classifiers and acoustic feature sets when classifying individual signatures.

## Leakage control: Some limits with a SUNG dataset

In the previous sections, we assessed whether each call bears enough idiosyncratic information to automatically attribute it to the individual who produced it. Similarly, we evaluated to which extent a classifier can tell one call type from the others. In both tasks, each call appeared either in the training or the test subsets thus guaranteeing the absence of subset overlap at the *call* level. We next evaluated the impact of the leakage between calls produced in the same *sequence*. Despite this, we do not achieve full independence between the subsets, as illustrated in the following.

First considering the classification of call types, classifiers would ideally be trained with a large amount of data recorded from many individuals. The evaluation would then consist in testing whether these classifiers correctly generalize their decision on calls produced by other individuals, distinct from the ones present in the training set but belonging to the same group to avoid potential dialectal differences. Such an experimental design is nevertheless beyond the reach of a SUNG dataset due to limited data availability.

Considering individual signatures, the same animals logically appear in both the training and test subsets but one may, however, improve their disjunction by assigning *distinct* call types to each subset. In such a configuration controlled for call types, classification decisions are necessarily based on individual-invariant characteristics rather than on a potential transfer of type-specific information from the training to the test subset.

To explore this option, given the imbalance in our initial dataset–with for instance no SCB from Kumbuka and only five SB from Busira, see Table 1 –we built a reduced dataset where each combination of type and individual is represented by at least 18 calls. This selection kept 58% of the original dataset and resulted in the breakdown between five individuals and four

types shown in Fig 14 (top panel). From this subset, one can draw several training/test repartitions that guarantee that for a given individual, the call types differ across the sets. For instance, in Configuration Example #1 shown in the middle panel of Fig 14, the classifier is trained on Peeps, Peep-Yelps, and Soft Barks produced by Djanoa and Zuani and their signature is tested on their Barks (unseen by the classifier during the training phase). For the three other individuals, the repartition differs but the same disjunction principle applies to control for call types. Configuration Example #2 (Fig 14, bottom panel) shows another possible split of the types and individuals. This procedure also generates much less evenly balanced splits, for instance with one type being represented by a single individual in the training set, and present in the test set for all the other individuals.

It is worth noting that these kinds of configurations do not reflect the daily situation experienced by the animals. More precisely, the fact that the bonobo repertoire has a limited number of call types implies that it is extremely uncommon for a bonobo to hear a social partner producing a call of a type never heard from this specific animal before. This approach, however, offers an interesting stress test for the classifiers in a SUNG context by providing the maximum disjunction between the respective type and individual characteristics.

We performed these additional classification experiments (for the individual signature task) with the same classifier architectures as in previous sections, with two main results. First, the performances are largely degraded albeit also essentially better than chance. Our interpretation is that an individual signature consists both in traits that can be extracted from the voice (and thus accessible to the classifier in this procedure) and from idiosyncratic ways of producing each type (not accessible to the classifier), as also suggested by our study of across-individual variability in Section I. Secondly, the variation in performance across the training/test split configurations is much larger than what we observed in our princeps experiment. In this particularly challenging setting, this instability suggests that the limited size of the dataset, coupled with the graded nature of the bonobo repertoire, does not allow for the training of classifiers that are robust enough to abstract from individual effects (which call type produced by which individual is in the training or the test sets). The results are provided in Supplementary Information (S2 Text).

## V. Discussion: Main achievements and limitations

While improving our understanding of how information is encoded in the bonobo vocal repertoire, this paper has primarily a methodological objective. Our goal is to assess the suitability of different approaches to vocalization processing (acoustic analysis, data visualization, and automatic classification) to describe information encoded in vocalizations from a SUNG dataset representative of what is commonly obtained in bioacoustic studies (Fig 1). Our approach consists of an acoustic characterization in several feature spaces from which a graphical representation is derived through S-UMAP, followed by the implementation of several classification algorithms and their combination to explore the structure and variation of bonobo calls (Fig 6) and the robustness of the individual signature they encode (Fig 9). We assess the importance of the notion of data leakage in the evaluation of classification performance, and the need to take it into account despite the inherent challenges (Figs 12, 13 and 14).

### Main findings

**Features describing acoustic signals.**   We compared a set of 20 acoustic parameters traditionally used to describe primate vocalizations (Bioacoustic set) with a reduced set of seven semi-automatically computed parameters (DCT set) and a more comprehensive spectro-temporal parameterization comprising 192 parameters (MFCC set). While traditional acoustic

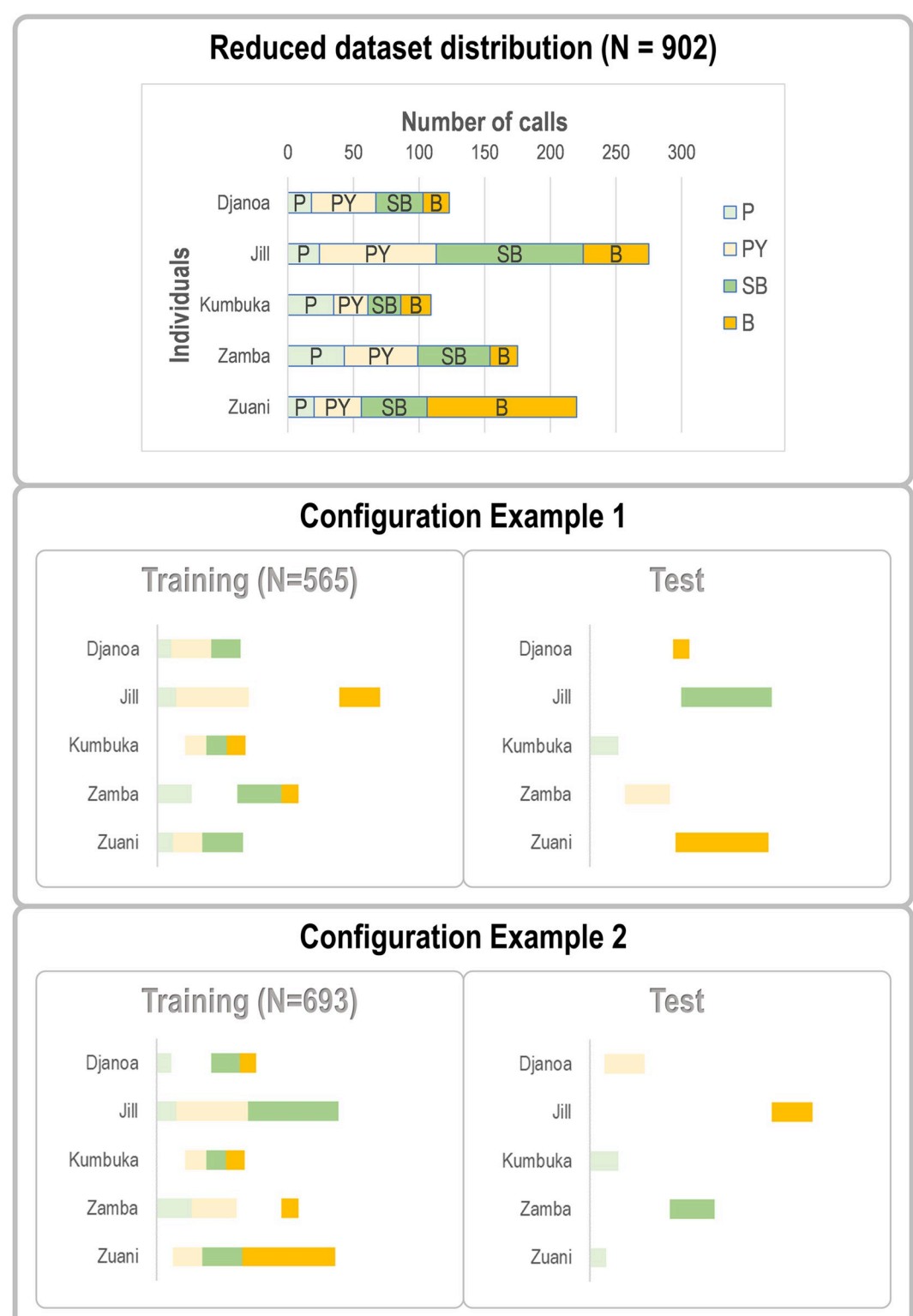

**Fig 14. Illustration of a strategy to minimize information leakage when building training and test sets for a classification / discrimination task.** The upper panel shows the distribution of call types per individual in our reduced dataset. The middle and lower panels display two configurations for the training and test sets where each call type for a given individual appears only in one of the two sets.

parameters appear adequate for characterizing call types, it appears that classifiers benefit from a finer-grained acoustic description (such as that provided by the MFCC) when characterizing individual signatures. This is probably because the MFCC parameterization does not make any assumptions about the relative salience of specific spectral or temporal parameters [3] and encodes subtle differences related to the emitter's anatomy and physiology beyond the overall basic characteristics of the calls, on the full spectrum. These parameters are sufficiently idiosyncratic to suggest that an ape can correctly identify the emitter in a real-life situation [92,93]. An algorithm-based automatic classification is, of course, only partially related to the cognitive task an animal performs when it has to decode the information carried by a conspecific's vocalization [94,95], for two main results. First, thanks to its social experience, each ape has built good priors and expectations on who can vocalize when and how. Additionally, in a real situation, a call is likely enhanced with contextual information that could involve visual cues and recent information on the previous moves and positions of the group members. These factors probably result in a simplified task, and it can be assumed that individual conspecific individual vocal identification is quite accurate in bonobos. This has been shown in an experimental study using playbacks with a single call type, namely the peep-yelp [53].

The fact that the performance reached with the DCT set on call type identification is only slightly worse than the best performance suggests that this representation adequately captures their essential aspects. By design, the first DCT coefficients characterize a call tonal trajectory in terms of average pitch (DCT0), global slope (DCT1), curvature (DCT2), while higher-order coefficients encode more rapid changes in the call. As a consequence, the coefficients' sign and magnitude may be linked to visual–and thus more intuitive–characteristics on a spectrogram. This is very encouraging as it can be expected that automatic feature extraction can be rapidly developed for bonobo (and probably other mammals) calls, also taking advantage of recent improvements in acoustic pre-processing [25,47,96]. Adopting multiple feature sets may also be good practice for extracting non-redundant information, particularly in small datasets. The very nature of the features themselves may obviously vary from one species to another. On the one hand, the Bioacoustic and DCT sets that we implemented are essentially aimed at characterizing the f0 shape and therefore at describing tonal signals. They are consequently well adapted to most of the bonobo repertoire. On the other hand, MFCC and related spectro-temporal representations are able to deal with both tonal and non-tonal calls, providing a quite generic feature space that can be adapted to a large variety of acoustic communication systems. The complementarity between expert predefined acoustic features (Bioacoustic or DCT set) and agnostic spectro-temporal representation (MFCC set) echoes the conclusions of [33] in the context of unsupervised visualization of a vocal repertoire.

**Visualization.**   Whether using illustrative raw spectrograms–as in [35] and [36]–or aggregated templates (as in this paper in Figs 3 and 4), graphical representations of vocalizations are essential to capture the structure and characteristics of any vocal communication system. The projection of a dataset into a feature space provides additional information about its variability and clustering. Recent nonlinear reductions and projections such as UMAP are particularly convenient and powerful to explore and discover the structure of large datasets, as convincingly stated by [33]. However, these authors acknowledged that in the context of small datasets, unsupervised UMAP representations may fail to clarify their structure (but see also [34]). Here, we overcome this limitation by showing that a supervised UMAP provides an elegant means of transforming our bonobo SUNG dataset from its acoustic description into an informative, parsimonious and discriminative latent feature space informed by the hand-labeled categories. This representation can in turn be used as an indication of the data clumpiness, the consistency of the labels (whether call types or individuals), and the resulting discriminant potential. Although we didn't detail this aspect in the present paper, it is worth mentioning

that such a representation can also help diagnose each datapoint and detect potential mislabelling, especially by using an interactive plot as the ones provided in the file '6_Analysis_SUMAP_and_Silhouette.html' in the Github repository.

**Automatic classification.**   We implemented three classifiers (svm, xgboost, and NN) and compared their performance to that of discriminant analysis (DFA), a method classically used to analyze the information carried by animal vocalizations, in particular vocal signatures. Our results show that all three models significantly outperform DFA for the identification of both call types and individual signatures. By design, DFA is based on linear decision boundaries and may prove to be insufficiently discriminant to tackle classification tasks involving nonlinear separation between classes (see S3 Text for an illustration of the differences between linear vs. nonlinear decision boundaries). DFA may thus miss the identification of acoustic categories and therefore not be the most adequate means to assess the categorical nature of a complex vocal repertoire or the accuracy of vocal signatures. Furthermore, the performance achieved when evaluating individual signatures (illustrated by the confusion matrix in Fig 8) shows a striking contrast between individuals for which the dataset is very limited (around ~75 calls for each of the three least represented individuals) and the most represented individuals (~200 calls or more for three individuals). This difference reminds us that we have processed a very small dataset (Table 1) by the standards usually expected for machine learning. Moderately increasing the dataset size by only a few hundred observations per individual would certainly bring a significant improvement in the performance of the classifiers, as also suggested by the better overall performances reached for call type classification, for which we had at least 200 calls in each category.

From a data scientist's point of view, it is also interesting to note that ensembling methods improve performance. This means firstly that the three feature sets encode complementary information to some extent. Furthermore, the three classifiers exploit them in different ways. Obviously, bioacousticians are not necessarily obsessed with achieving the best classification performance. In fact, a trade-off between performance and interpretability (transparent and explainable features being preferable to opaque parameterization) must often be sought, and prohibitive computation time may also be an obstacle. The general picture drawn from our experiences is that a neural network, though better than DFA, is more challenging to deploy than svm and xgboost because of the technical requirements of the overall deep learning framework, without outperforming those simpler approaches. In contrast, both svm and xgboost perform quite well and reach very similar results in both tasks (call type and individual recognition) with an adequate feature set. High-accuracy classifiers, such as a carefully optimized xgboost or a stacked learner made of svm and xgboost for instance, could be very useful when high accuracy is desired, for example when examining a new vocal repertoire, or for passive acoustic monitoring of wild animals ([97–99] for recent overviews). They would also be the best choice to automatically classify unlabeled calls and increase the dataset size, in a process known as iterative pseudo-labeling [100]. Their downside is nevertheless that optimizing such models relies on exploring a complex hyperparameter space and consequently requires some solid expertise in machine learning. Even if xgboost seems to have an edge on svm, we thus suggest that a simple classifier such as svm offers an adequate trade-off, being sufficiently informative, simpler to implement and much less computer-intensive than xgboost for instance (especially when tuning the different hyperparameters and in the repeated procedure necessary for a well-controlled cross-validation framework).

**Performance assessment.**   Performance assessment must be done with great care. Dealing with a SUNG dataset means taking into account data sparsity, class imbalance, and potential data leakage. Their detrimental effects can be mitigated to some extent, by adopting the right metrics and models, and by carefully controlling for leakage induced by confounding factors

between the training and test sets. The high level of robustness observed in speech or speaker recognition with very large corpora (based on thousands of speakers and various recording conditions, to mention only two aspects) is unreachable with relatively small datasets. Even if one carefully assigns calls recorded at different times to the training and test sets, there is inherently some residual data leakage when assessing the individual signature, as the soundscape, recording conditions and equipment often vary between recordings. These caveats make it difficult to compare approaches and papers. In this regard, multicentric challenges offering a unified evaluation framework based on a shared dataset should be encouraged [47].

## Generalization and recommendations

Beyond the thorough quantitative characterization of the variability observed in the vocal repertoire of a group of bonobos and of the vocal signature that can be inferred from their calls, our results lead to the identification of several practical approaches that are generalizable to any other animal communication system. We suggest that adopting them can improve the standard bioacoustics workflow in terms of information gain and reproducibility, especially with SUNG datasets. These approaches may probably seem standard to some machine learning experts [25], but the current literature shows that they are not yet adopted in a systematic way. However, this is not a magic recipe, and some thought should be given to whether these recommendations are relevant to the context of interest.

We recommend:

1. comparing several acoustic parameterizations as they may be complementary;

2. visualizing the dataset with Supervised UMAP to examine the species acoustic space;

3. adopting SVM (Support Vector Machines) rather than discriminant functional analysis as the baseline classification approach;

4. explicitly evaluating data leakage and possibly implementing a mitigation strategy.

Paradoxically, an additional suggestion would be not to rely too much on the SUNG corpora and to remain cautious in their interpretation. No matter how impressive the advances in machine learning are, collecting large amounts of high-quality recordings will remain the only way to get a comprehensive and robust picture of an animal communication system.

## Future work

The approaches implemented in this study offer a reasonable balance between performance and complexity. Several recent approaches offer interesting alternatives or potential improvements. The direct use of spectrograms, either as an image or as a parameter matrix, has already been applied to mice [44], Atlantic spotted dolphins [101], domestic cats [102], common marmosets [52], etc. However, their performance on complex and graded repertoires remains to be evaluated, and adaptation of spectrogram parameters to each species may be necessary [82,96] have recently fine-tuned a deep learning model trained on human voice to detect chimpanzee vocal signatures. This procedure is promising because conceptually it should work, but the results need to be confirmed on a larger task with stringent control of data leakage. Other neural network architectures have also been successfully implemented on large datasets (e.g., [103] on mouse ultrasonic vocalizations). Progress has also been achieved in segmenting long audio bouts into calls [52,104,105], but the proposed solutions currently struggle with the adverse conditions typical of SUNG datasets (overlapping calls, non-stationary noise, etc.). All these approaches are nevertheless expected to improve classification accuracy on large-enough datasets, but the main performance limitation comes from the dataset itself. As

aforementioned, its low audio quality may partially be compensated by an adequate acoustic preprocessing, but complementary approaches lie in its size artificial extension by data augmentation techniques [106] or by developing multi-species models or transfer learning approaches [107]. These techniques are not magic bullets, though, and their efficient implementation will require additional studies [44].

Finally, the automatic analysis approaches discussed in this paper provide efficient tools to help bioacousticians tackle a vocal repertoire. The proposed workflow addresses an intermediate, but essential, phase on the way to a global understanding of an animal communication system, and it thus takes part in a collective effort to adapt recent machine learning approaches to bioacoustics.

## VI. Detailed methodology

### Ethics statement

All research conducted for this article was observational. All data collection protocols were performed in accordance with the relevant guidelines and regulations and were approved by the Institutional Animal Ethical Committee of the University of Saint-Étienne, under the authorization no. 42-218-0901-38 SV 09.

### Extraction of acoustic features

**Bioacoustics**: this set of acoustic parameters is inspired by the procedure described in [37]. It consists of parameters computed with Praat (version 6.0.24, from May 2017) that summarize the $f_0$ shape and the energy distribution for each call. The call harmonicity (or Harmonics-to-Noise Ratio, HNR), computed with Praat has also been included. Associated with the perception of *roughness*, HNR measures the ratio between periodic and non-periodic components of an acoustic signal. If HNR, expressed in dB, is close to 20 dB, the vocalization is considered mostly periodic. If HNR is rather close to 0 dB means that there is equal energy in its harmonic and noise components. This procedure results in 20 features. *Manual operations required*: *call segmentation and $f_0$-Peak positioning*.

**MFCC:** this set computed with the voicebox toolbox in Matlab, downloaded from https://github.com/ImperialCollegeLondon/sap-voicebox in November 2019, is developed from a Mel-frequency cepstral analysis computed through 32 triangular shaped filters distributed over the 500–12000 Hz band. The coefficients are computed for successive frames (~23 ms duration and 50% overlap) and a Hamming window is applied. The command v_melcepst(S, Fs,'dD',32,33,1024,512,500/Fs,12000/Fs) is applied to the audio signal S, sampled at frequency Fs. First and second order derivatives (so-called delta and delta-delta coefficients) are also computed, resulting in 3 x 32 = 96 dimensions. The final MFCC set consists of the average and standard deviation computed over the call. This procedure results in 192 features. *Manual operation required*: *call segmentation*.

**DCT**: this set is computed with Praat and is mainly based on the $f_0$ contour, on which a Discrete Cosine Transform (DCT) is applied. DCT is a method used in phonetics to parametrize the contours of the fundamental frequency ($f_0$) and formants [63, 64, 108, 109]. This parameterization decomposes the signal into a set of values that are cosine amplitudes of increasing frequency that compose the $f_0$ value sequence between the onset and the offset of the vocalization. Each coefficient characterizes an aspect of the shape of the $f_0$ trajectory relative to a cosine. The zeroth DCT coefficient is a value proportional to the mean of the original $f_0$ trajectory; the first coefficient is equivalent to the magnitude and direction of change from the mean; the second is related to its trajectory's curvature. The higher order coefficients represent the amplitudes of the higher frequency cosines and thus correspond to increasingly detailed

information about the $f_0$ trajectory's shape. In the following analysis, the $f_0$ contours have been approximated by the first five DCT coefficients. The call duration and harmonicity were also included. The formula used in a Praat script to compute the DCT coefficients was from [63]. This procedure results in 7 features (5 DCT coefficients + Duration + Harmonicity). *Manual operations required*: *call segmentation and $f_0$ contour checking.*

All analyses detailed in the following methodological presentation were implemented with the R software [110]. A detailed list of the packages is provided at the end of the section.

## Data selection and preprocessing

We selected a subset of individuals and call types from our initial dataset, specifically the five types P (peep), PY (peep-yelp), SB (soft bark), B (bark), and SCB (scream bark), and individuals with more than 70 calls in these five types, namely Bolombo, Busira, Djanoa, Hortense, Jill, Kumbuka, Lina, Vifijo, Zamba, and Zuani. This selection resulted in a data set of 1,560 observations (S1 Data).

Only one missing value was found in a bark produced by Jill. The $f_0$ value in the center of the vocalization (f0.mid) was not detectable. We could have chosen to simply delete this occurrence, but in order to provide a more generic solution for datasets with a larger number of missing values, we used a random forest algorithm. This algorithm is trained on the observed values of the variables related to $f_0$ (q1f, q2f, q3f, f.max, q1t, q2t, q3t, t.max, f0.sta, f0.mid, f0.end, f0.av, f0.max, tf0.max, f0.slope.asc, f0.slope.desc, f0.slope.1st.half, f0.slope.2nd.half, f0.onset, f0.offset) in order to offer a plausible reconstruction.

Prior to the application of the classifiers, the different sets of predictors were standardized by centering them at the mean values and rescaling them using the standard deviation (z-scoring).

## Imbalance in the data

Our dataset is characterized by an imbalance in the way our observations—the calls—are distributed across individuals and call types (not to mention other dimensions such as the context of emission). Table 1 summarizes the situation.

While the most represented individual—Jill—has 362 calls, the least represented individuals —Busira and Bolombo—have about 20% of this amount. The call types are more evenly distributed, with the number of occurrences ranging from 206 to 443. It is important to note that some individuals lack certain call types, e.g., Kumbuka and SCB, or have only a few occurrences, e.g., six B for Bolombo or five SB for Busira.

Such imbalance is a common feature of SUNG datasets and should be taken into account when considering classification approaches. Although permuted DFA is inherently designed to deal with this type of imbalance, we considered several options for the other approaches. Results are reported with class weights inversely proportional to the number of observations in each class, which yielded better results than undersampling, oversampling or SMOTE (synthetic minority oversampling technique).

It should be noted that we only paid attention to the imbalance for the target domain of classification, i.e., when classifying individuals, we only considered the different numbers of calls per individual and not for the call types, and vice versa.

## Supervised UMAP and silhouette scores

To compute supervised UMAP representations, we used the *umap()* function of the *uwot* package with the following settings: 100 neighbors, a minimal distance of 0.01 (the default value), and two dimensions. The Euclidean distance was used to find nearest neighbors. The

target labels (call type or individual signature) were also passed as parameters to perform the supervised dimension reduction (S-UMAP). Specifically, the hyperparameters *n_neighbors* and *min_dist* are used to control the balance between local and global structure in the final projection [111]. *n_neighbors* is the number of neighboring points (for each data point) used to construct the initial high-dimensional graph. This means that a high value such as 100 forces the algorithm to concentrate on the very broad structure (at the loss of the more detailed local structure). *min_dist* is the minimum distance between points in the final low-dimensional space. It controls the extent to which UMAP aggregates points, with low values resulting in more clumped embeddings [84].

For the silhouette scores (silhouette widths), we used the get_*silhouette()* function of the *clues* package. Here, the silhouette scores are computed with the default Euclidean distance metric.

## DFA

In the first step of the discriminant analysis, a training sample was used to compute a set of linear discriminant functions. In order to test for call type distinctiveness, the training sample was randomly selected from each call type without controlling for the individual. Similarly, in order to test the individual vocal signature, the training sample was randomly selected from each individual's vocalizations. The number of occurrences selected was the same for each call type and each individual. This number was equal to two thirds of the smallest number of vocalizations analyzed per call type (i.e., 206×2/3) and per individual (i.e., 71×2/3). These discriminant analysis models computed from training samples were used to classify all remaining occurrences. Thus, for each individual and for each type of vocalization, at least one third of the occurrences were included in the test sample. We performed 100 iterations of these discriminant analyses with randomly selected training and test samples. The performance metrics and the percentage of correct classification were obtained by averaging the results obtained for each of the 100 test samples.

In contrast to the SOTA approaches mentioned below, discriminant analysis does not automatically handle multicollinearity between predictors, which must be explicitly addressed. We chose to identify and reduce collinearity in the bioacoustic and DTC sets by computing the variance inflation factor (VIF). As suggested by [112], we used a stepwise strategy. The stepwise VIF function we used was written by Marcus W. Beck and was downloaded from https://gist.github.com/fawda123/4717702. The VIF values for all predictors were computed, and if they were larger than a preselected threshold (here, 5), we sequentially dropped the predictors with the largest VIF, recalculated the VIFs and repeated this process until all VIFs were below the preselected threshold. A tick in S1 Table indicates the predictors included in DFA analyses. Regarding MFCC, out of the 192 features, only the first 96 corresponding to the 32 standardized coefficients and their first and second derivatives were used, as the standardized values of standard deviations were considered as constants by the DFA algorithm. To reduce the number of parameters used as input, but also to handle multicollinearity between predictors, a principal component analysis (PCA) was used as a front-end [41]. The optimal number of principal components (PC) was determined by training the classifier with a varying number of PC coefficients and by estimating each time the performance of this classifier using the same cross-validation technique mentioned above in this section (see Fig 15).

Given the use of log loss to tune the hyperparameters of our SOTA models, the number of PCs leading to the minimal log loss was chosen as optimal. More specifically, we considered a Loess regression (Locally Estimated Scatterplot Smoothing) as a smoother and chose the minimum of the regression curve. In a Loess regression, the fitting at say point *x* is weighted toward

the data nearest to $x$. The distance from $x$ that is considered near to it is controlled by the span setting, $\alpha$. If the span parameter is too small, there will be insufficient data near x for an accurate fit. If it is too large, the regression will be over-smoothed. Span values from 0.25 to 10 with increments of 0.25 were tested for call types and individual signatures. 2.5 was eventually chosen for both–values higher than this threshold lead to the same coordinates for the minimum of the regression curve. In the case of individual signatures, this threshold corresponds to 38 PCs and in the case of call types, 41 PCs. These PCs explain respectively 82% and 84% of the overall variance of the standardized MFCC and their first and second derivatives. The DFA results were thus obtained by describing each call with the coefficients of these 38 or 41 principal components respectively.

## Training/testing procedure

When working with a large dataset and considering a supervised ML technique, it is common to split it once between a training set and a test set. However, as the size of the dataset decreases, however, this binary sampling procedure gives rise to an increasing risk of sampling error. In other words, the results may depend significantly on the specific observations falling randomly into the two sets, as some observations are inherently more difficult to classify than others, a well-known issue in the domain of human speaker recognition [113]. It is therefore not prudent to assess performance with a single split. One should note that this issue is orthogonal to exerting some control over the two sets as what we did to ensure similar representativity of the classes of observations in both train and test the two sets (e.g., when classifying individuals, 23% (362/1560) of the calls should be Jill's in both sets). To account for the previous risk, we averaged performances over 100 repetitions, i.e., over 100 pairs of training and test sets, each time with an 80%-20% split. The performance distributions clearly illustrate the effect and range of sampling error, and the overall symmetry of these distributions validates the use of mean values as an estimator of central tendency. We computed mean values not only for our different metrics (see Section II), but also for the confusion matrices and to assess feature importance.

## Hyperparameters for SVM, NN and XGBoost

We considered the following hyperparameters for our different 'state-of-the-art' classifiers (with corresponding possible value ranges):

For svm: i) the nature of the kernel (linear, polynomial or radial), ii) for the polynomial kernel, the number of degrees (between 1 and 4), iii) the cost parameter C, which is a regularization parameter that controls the bias–variance trade-off (values between $2^{-7}$ and $2^{7}$, with a power-2 transformation), and iv) for the radial kernel, the parameter gamma or sigma, which determines how the decision boundary flexes to account for the training observations (values between $2^{-13}$ and $2^{5}$ with a power-2 transformation).

For NN, there are quite a number of parameters, involving for instance different weight initialization schemes (glorot normal, he normal, glorot uniform etc.), the optimizer (rmsprop, adam etc.) or regularization techniques like dropout, and we only focused on some of them: i) the number of epochs, i.e., of training periods (between 25 and 200), ii) the number of layers before the final layer (between 1 and 3), iii) the number of neurons in the first layer (between 5 and 100), iv) if relevant, the number of neurons in the second layer (between 5 and 100), v) if relevant, the number of neurons in the third layer (between 5 and 100), vi) the drop-out rate for each of the previous layer (one value shared by all layers, between 0.1 and 0.5) and vii) the input drop-out rate for—as the name suggests—the inputs provided to the initial layer (between 0.1 and 0.5). The final layer contained a number of units equal to the number of

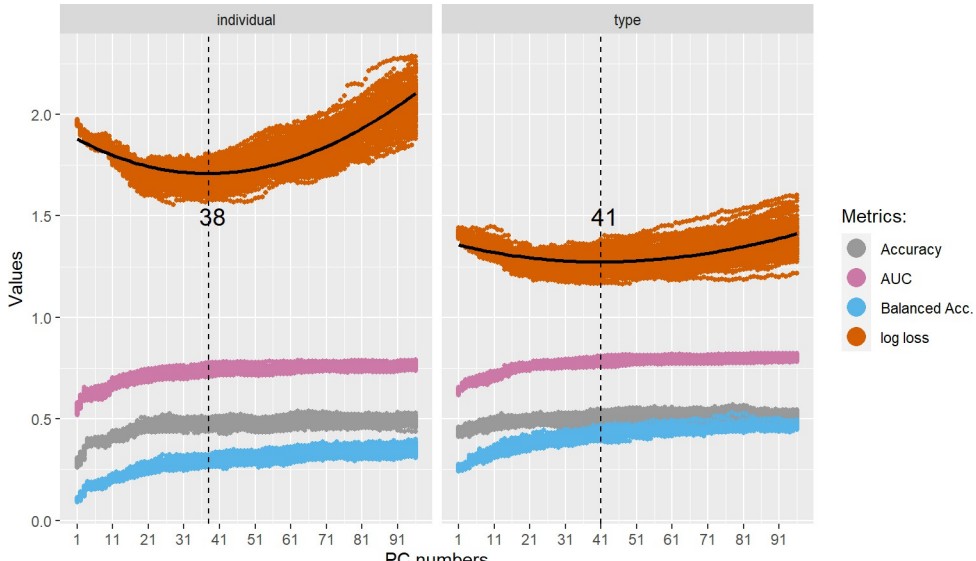

**Fig 15. Scatterplots showing the evolution of the DFA performance as a function of the number of PC considered using four different performance metrics (see "Automatic classification approaches and evaluation methodology" for details about these metrics). Left.** Classification of individual signature. **Right.** Classification of call type.

classes of the classification problem. A softmax activation was used for this layer in relation to a cross-entropy loss function. We choose a size of 128 for the mini-batches, a glorot uniform weight initialization scheme, and an Adam optimizer with a learning rate of 0.001.

For xgboost, we focused on 8 parameters and chose default values for others for the sake of simplicity: i) the maximum number of rounds/iterations, which for classification is similar to the number of trees to grow (between 10 and 1000), ii) the learning rate eta, which shrinks the feature weights after every round, and for which low values slow the learning process down and need to be compensated by an increase in the number of rounds (between 0.01 and 0.2), iii) the regularization parameter gamma, which relies on across-tree information and usually brings improvement with shallow trees (values between 2^-20 and 2^6 with a power-2 transformation), iv) the maximal depth of the tree—the deeper the tree, the more complex problems it can handle, but with a higher risk of overfitting (values between 1 and 10), v) the minimum 'child weight', which corresponds to the minimum sum of instance weight of a leaf node (calculated by second order partial derivative), and helps decide when to stop splitting a tree and block potential feature interactions and related overfitting (between 1 and 10), vi) the number of observations supplied to a tree (values between 50% and 100%), vii) the number of features supplied to a tree (values between 50% and 100%), and viii) the regularization parameter alpha, which performs L1 (lasso) regression on leaf weights (values between $2^{-20}$ and $2^6$ with a power-2 transformation). Hyperparameter tuning was the most computationally demanding for xgboost because we chose to work with large ranges of values, which in some extreme cases required a very large number of calculations.

**Tuning procedure.** There are different methods for exploring a space of hyperparameters [114]. First, a random search can be used, in which a number of acceptable configurations of hyperparameter values, chosen at random, are tried and compared. The number of configurations must increase, however, as the number of hyperparameters increases and the possible ranges of values widens. In our case, this approach did not appear to be the most efficient. Another option is to perform a grid search, where the range of possible values for each hyperparameter leads to a number of evenly spaced values (possibly with a log/power

transformation), and all sets of these values are tried and compared. Again, this method quickly becomes intractable as the number of hyperparameters increases and one wishes to consider a dense grid. A third option is to perform a tailored search when it makes sense to consider some hyperparameters first and then others, thus greatly reducing the number of configurations to be explored. Such an approach can be found for instance for xgboost, but not for other techniques. A fourth approach is model-based optimization (MBO, also known as Bayesian optimization), in which a probabilistic model gradually learns the structure of the hyperparameter space and which hyperparameter values lead to the best performance [115]. This approach is generally much less computationally intensive than previous approaches. A multi-point approach can be considered especially when parallelization of the computations is possible.

We took advantage of a multi-point random-forest-based Bayesian MBO procedure with 25 iterations using a "lower confidence bound" infill criterion [116], which is suitable for a parameter space that is not purely numerical. We indeed compared this approach with the previous ones and found that relying on a model-based optimization led to the best ratio of performances over time. For each 'state-of-the-art' classifier and each set of predictors, we parallelized the procedure.

Given our 100 replicated training-test splits, in conjunction with our different algorithms and predictor sets, it would have been extremely time-consuming to perform an hyperparameter optimization for each configuration. We therefore decided to separate the hyperparameter tuning from our assessment of classification performances. For each classifier and each set of predictors, we performed a 5-time repeated 5-fold cross-validation to assess the hyperparameters. We assume that this provides enough cases, i.e., enough different validation sets, to compute values of the hyperparameters that perform well on average over many different configurations—our 100 repetitions.

**Performance metrics for tuning the hyperparameters.**   A metric was required for the tuning of hyperparameters. We considered log loss both because it is the only metric in our set of metrics that can be used to train neural networks, and because at a more theoretical level, it corresponds to a 'pure' Machine Learning perspective, independent from the 'real problem' (bonobos calls)—log loss as a metric tells us whether the model is confident or not when making a prediction.

As a check, we also used AUC to tune the SVM and xgboost hyperparameters, and found very similar results to those obtained with log loss.

## Feature importance

While some algorithms such as xgboost have their own specific techniques for estimating feature importance, we chose a generic method that can be applied to any of our three algorithms SVM, NN and xgboost. It consists in considering each feature in the feature set independently, and for each of them, comparing the quality of the predictions obtained with the initial values with the quality of the predictions obtained after randomly permuting these values across observations. Intuitively, the lower the performance when shuffling, the more important the feature and its values are to classify the observations. This corresponds to the implementation, with the method 'permutation.importance', of the function ***generateFeatureImportanceData()*** in the **mlr** package [84]. We chose a number of 50 random permutations to avoid extreme results obtained by chance.

We considered feature importance for both Bioacoustic features and DCT, but left out MFCC due to the number of variables and the difficulty of assigning individual articulatory significance to them.

Again, since we ran 100 iterations for each algorithm and feature set configuration, we averaged the values of feature importance for these interactions to neutralize the sampling error of any given iteration.

### Ensembling

We considered seven different stacked ensembles:

– For each classifier (SVM, NN and xgboost), we first stacked the models corresponding to the three different feature sets (Bioacoustic parameters, DCT coefficients and MFCC coefficients).

– For each feature set (Bioacoustic, DCT, MFCC), we conversely stacked the models corresponding to the three 'state-of-the-art' classifiers.

– Finally, we stacked the 9 models corresponding to the three feature sets processed by the three classifiers.

As for the super learner, we considered a penalized multinomial regression with an L2 penalty (ridge regression) [117].

### Evaluating data leakage / Bali

The Default strategy was implemented through a random sampling procedure which only controlled for equal proportions of the classes of observations in the training and test set. The Fair and Skewed strategies, on the other hand, were implemented with an in-house tool called Bali. Based on the object-oriented R6 framework in R software, Bali relies on a genetic algorithm (with only random mutations and no genome recombination) and the definition of a number of rules—with possibly different weights—to be respected when creating sets of specified sizes. Some rules are for continuous variables and aim to equalize or differentiate the mean or variance across rows or columns of different sets. Others deal with categorical variables, and aim to maximize or minimize diversity, etc. Bali is currently in development and not publicly distributed. The final version will be released on github under a Creative Commons license soon and cross-referenced in http://github.com/keruiduo/SupplMatBonobos.

We first defined a rule to ensure similar distributions of occurrences of individuals in the training and test sets. When only this rule is given, the algorithm consistently achieves the objective, with a result similar to the Default strategy. By adding a second rule to prevent or maximize the overlap of sequences on the two sets, we implemented the Fair and Skewed strategies respectively.

### A step-by-step demo

As we recommend that SVM be adopted as the baseline classification approach, a step-by-step demo in which SVM is used to predict individual signatures with a small set of acoustic features consisting of duration, HNR and 5 DCT coefficients is provided in the file '7_Example_mlr_svm.html' available in the Github repository (http://github.com/keruiduo/SupplMatBonobos). The code includes the computation and interactive display of a supervised UMAP.

### Implementation

We relied on the following R packages:

– for the generic ML functions and procedures, *caret* [118] for DFA and *mlr* [84, 119] for SV, NN, xgboost and the ensemble learners

– *MASS* for the DFA [120]

– *mclust* for the gmm [121]

– *splitstackshape* to prepare sets with identical proportions of classes of objects [122]

– *mlrMBO* for the model-based optimization of the hyperparameters [116]

– for parallelization of the 100 repetitions to assess performances, *parallelMap* [123]

– **keras** for the neural networks (CPU-based version 2.8, Python 3.9) [124]

– *ggplot2* for the various figures [125]

– *uwot* for the umap [126] and *clues* for the calculation of the silhouette scores [127]

– for data processing broadly speaking, **tidyverse** [84, 128]

– *R6* for the BaLi algorithm [129]

– *missForest* to impute missing values [130]

– *Plotly* to create interactive S-UMAP plots [131]

The code made available to reproduce the experiments detailed in this paper has been tested and run on a personal computer (AMD Ryzen 9 5950X 16-core processor, 32GB RAM), with parallelization for i) the hyperparameter tuning of SVM, NN and xgboost with repeated cross-validation, ii) the estimation of the classification performance for simple and stacked learners with 100 runs, and iii) the estimation of the random baseline with 1,000 runs.

## Supporting information

**S1 Data. Raw dataset.**
(TXT)

**S1 Fig. Examples of spectrograms of the five call types.**
(TIFF)

**S1 Table. Description of the acoustic features and feature sets.**
(PDF)

**S1 Text. Comparison of the classification performances of SVM with different MFCC sets.**
(PDF)

**S2 Text. Controlling for call types in the individual signature task.**
(PDF)

**S3 Text. Illustration of the difference between linear and nonlinear decision boundaries.**
(PDF)

**S4 Text. Description of the sound recording provided in S1 Sound.**
(TXT)

**S1 Sound. Example of sound recording (Call # 698 by Jill).**
(WAV)

## Acknowledgments

We warmly thank the zoological parks of Apenheul, Planckendael and La Vallée des Singes for their welcome, and especially the bonobo keepers for their support.

## Author Contributions

**Conceptualization:** Vincent Arnaud, François Pellegrino, Sumir Keenan, Xavier St-Gelais, Nicolas Mathevon, Florence Levréro, Christophe Coupé.

**Data curation:** Sumir Keenan, Xavier St-Gelais, Florence Levréro.

**Formal analysis:** Vincent Arnaud, François Pellegrino, Xavier St-Gelais, Christophe Coupé.

**Funding acquisition:** Vincent Arnaud, François Pellegrino, Xavier St-Gelais, Nicolas Mathevon, Florence Levréro.

**Investigation:** Vincent Arnaud, François Pellegrino, Sumir Keenan, Xavier St-Gelais, Nicolas Mathevon, Florence Levréro, Christophe Coupé.

**Methodology:** Vincent Arnaud, François Pellegrino, Sumir Keenan, Xavier St-Gelais, Nicolas Mathevon, Florence Levréro, Christophe Coupé.

**Project administration:** Vincent Arnaud, Florence Levréro.

**Resources:** Vincent Arnaud, François Pellegrino, Sumir Keenan, Xavier St-Gelais, Nicolas Mathevon, Florence Levréro, Christophe Coupé.

**Software:** Vincent Arnaud, Christophe Coupé.

**Supervision:** Vincent Arnaud, François Pellegrino, Florence Levréro, Christophe Coupé.

**Validation:** Vincent Arnaud, François Pellegrino, Sumir Keenan, Xavier St-Gelais, Nicolas Mathevon, Florence Levréro, Christophe Coupé.

**Visualization:** Vincent Arnaud, François Pellegrino, Florence Levréro, Christophe Coupé.

**Writing – original draft:** Vincent Arnaud, François Pellegrino, Christophe Coupé.

**Writing – review & editing:** Vincent Arnaud, François Pellegrino, Nicolas Mathevon, Florence Levréro, Christophe Coupé.

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
