## [Decision Letter · Decision Letter 0]

18 Aug 2022

Dear Dr. Coupé,

Thank you very much for submitting your manuscript "Improving the workflow to crack Small, Unbalanced, Noisy, but Genuine (SUNG) datasets in bioacoustics: the case of bonobo calls" for consideration at PLOS Computational Biology.

As with all papers reviewed by the journal, your manuscript was reviewed by members of the editorial board and by several independent reviewers. In light of the reviews (below this email), we would like to invite the resubmission of a significantly-revised version that takes into account the reviewers' comments.

Dear Arnaud et al,

As you see your paper has been reviewed by 3 expert referees. They all agree that your work represents a significant contribution to the field of bioacoustics and would fit well in Plos Comp Biology. They are also raised both major and minor concerns that should be addressed. The goals of the papers should be stated more clearly (and maybe in a more streamlined fashion) as suggested by Reviewer #3. You will also need to address the methodological concerns of Reviewer 1. I look forward to reading your revised manuscript.

Frédéric Theunissen

We cannot make any decision about publication until we have seen the revised manuscript and your response to the reviewers' comments. Your revised manuscript is also likely to be sent to reviewers for further evaluation.

Sincerely,

Frédéric E. Theunissen

Academic Editor

PLOS Computational Biology

Natalia Komarova

Section Editor

PLOS Computational Biology

Dear Arnaud et al,

As you see your paper has been reviewed by 3 expert referees. They all agree that your work represents a significant contribution to the field of bioacoustics and would fit well in Plos Comp Biology. They are also raised both major and minor concerns that should be addressed. The goals of the papers should be stated more clearly (and maybe in a more limited fashion) as suggested by Reviewer #3. You will also need to address the methodological concerns of Reviewer 1. I look forward to reading your revised manuscript.

Frédéric Theunissen

Reviewer's Responses to Questions

**Comments to the Authors:**

Reviewer #1: Overall, the manuscript provides a thorough overview of the very typical kind of bioacoustics datasets, where having the data "speak for itself" is not straightforward. I warmly welcome the methodological focus on this type of data, and especially the nuanced view put forward by the authors. Rather than claiming a single solution for all analyses, they outline a process, demonstrate it on a running example, and add the necessary caveats in their conclusions. Moreover, importantly, the whole manuscript and proposed methodology is supported by an extensive range of computational experiments, resulting in a complete, well-founded study. Pointing out the sometimes subtle and often ignored issue of data leakage is a very welcome contribution to the methodology of the field.

Overall, I have 3 major points I would like to see addressed in a revision:

1) Whereas the writing is clear to read, the overall structure and transitions between different sections and paragraphs could be improved.

* Most importantly, parts of the experimental setup described in Section VI are necessary to clearly understand the results, but are not forward-referenced. As an example, the number of repetitions averaged out and contributing to the uncertainty estimates of the classification techniques (pDFA and ML) are not clearly introduced in Section II and "suddenly" turn up. (See detailed notes for more examples, in context.)

* Especially in the first section/half of the manuscript, it should be made sure that it si more clear which statements deal with the general properties of SUNG datasets, and which ones are describing the specifics of the bonobo dataset. The idea of illustrating the abstract properties of SUNG datasets with a concrete example is great, but the separation of these two should at all points be clear to the reader.

* Some paragraphs have rather abrupt transitions, and obscure the overall line of reasoning (some examples noted further down).

=> In general, the manuscript feels as if it has been heavily restructured before. It would strongly benefit from a thorough front-to-back reading by the authors, keeping in mind the reader's knowledge and perspective up to that point, adding the necessary forward and backward references, and introducing a more details, but still intuitive, description of the experiments before the results.

2) The fourth and final part, visualising the data with UMAP and calculating silhouette scores, is not clearly reproducible and does not serve a valid purpose, as far as I can tell.

* Firstly, the purpose: while I am 100% convinced visualisation of datasets is crucial, I cannot find the goal of this extra analysis step *at the end* of the pipeline. The whole manuscript presents a very typical supervised machine learning methodology, having a statistical algorithm learn the human-assigned "ground truth" based on each data point's extracted features. So the extra, unsupervised clustering step at the end to find categories seems useless. I will assume that the authors did have a good reason, and would like to hear more about this, but in the current form, the article does not motivate this aproach.

* Please note that the referenced article by Sainsburg et al., where UMAP is being used, has an entirely different purpose (i.e., exploring call repertoires and finding call categories). On the other hand, this manuscript's methodology before section IV presupposes existing categories, determined beforehand. The additional benefit of UMAP (or other unsupervised/clustering methods) is lost to me, *especially* at the end of the analysis pipeline. A short discussion of the high-level differences in purpose and approach with Sainsburg et al. could clarify.

* It is also not clear how, in the 2nd and 3rd application of UMAP, the predictions are "included". Neither in section IV, neither in the details of section VI could I distill the exact details that I need to know to reproduce this approach (potentially, page 36 describes this partly, but it should be made much clearer).

* Finally, it seems obvious that including the predictions (which previous results showed were 50 to 80% accurate) would improve the clustering. Since the classes are reasonably well predicted by the supervised models, and these classes are discrete values, an increased quality clustering seems very unsurprising. Moreover, the value of that clustering seems to not add any insights about the data, as the clustering of well-predicted classes of the input data conforming to the same input data's classes is a rather circular result? It is however hard to judge as I cannot reconstruct the exact details of how these predictions were included in the UMAP clustering.

=> In general, the UMAP/visualiation section needs a big rework, clearly stating the intended purpose as well as how the input to the UMAP algorithm was constructed. In the current form, this section completely fails to convince me on both fronts, even though I would like to be and am generally hugely in favor of such visualisations approaches to better understand the structure of a dataset. This is an important issue to be resolved in order for this paper to be accepted and published.

3) As far as I can tell, XGBoost is a software libary, providing generic gradient boosting implementations which can be used in many different context. Based on the authors' description as well as XGBoost's online documentation (https://xgboost.readthedocs.io/), the actual algorithm/supervised models trained are "gradient boosted trees". Please dig into this, figure out the correct terminology, and correct the manuscript accordingly.

Minor comments and examples:

- p1, 22. "UMAPs": I believe "UMAP" as the name of the algorithm can just be used in singular here.

- p3, 2. "discriminate data samples": it could be useful to introduce the concept of "features" here, while describing the data points and training/test data.

- p3, 11-13. A short and/or intuitive way to describe the way pDFA mitigates this risk, and what remaining limitations it has, this would make this section a lot more useful. At the very least, a reference for the interested reader to find out more about the "its limitations" would be appreciated.

- p5. In general (not just Figure 1), the resolution of the images and figures is very low. I don't know if this is releated to PDF quality or the printer, in my case, but the authors should verify that the figures have a high enough resolution and sharpness.

- p5, caption. "Workflow implemented to analyze ...": Is this the proposed general workflow, the illustrated workflow with the bonobo data, or the workflow applied for this paper to set up experiments. Please clarify (and as stated before, try and keep the "general" case clearly distinct from the "specific" illustration with bonobo data).

- p6, 23. "shown in Figure 2": Looking forward to Figure 2, it does not contain a clear list of the call types (only scattered and duplicate abbreviations).

- p6, 34-35. "this kind of individual signature depending on the type of call": unclear what the authors are actually referring to. Please clarify/rephrase.

- p6, 36 - p7, 1. Example of a very abrupt transition. Also an example of jumping back and forth between SUNG-general and bonobo-specific sentences. In general, Section I and pages 6 and 7 could be improved b. making the links clearer between statements about bonobos and how this *relates* to typical issues in SUNG data.

- p7, 8-15: Great argument on the issue with small datasets and their caveats, and how bioacoustics data differs in important ways from speech!

- p8, 6 (and everywhere else). "F0": I've been told by reviewers before to use f0 (with 0 in subscript, ideally) as notation, and this does seem to be the more common notation. This also avoids confusion with formants.

- p9. However interesting the "Dataset distribution" is as visualisation, I am having a hard time intuitively reading and interpreting these areas. I am wondering if this is the most intuitive visualisation. E.g., what about either a heatmap (similar to Table 1, but colored by number), or stacked bars (one bar per animal, multiple segments per call type). Such visualisations would more easily allow comparison between the size of e.g. Vifijo and Djanoa (I have absolutely no idea which of the two has the most number of calls, because of their uncomparable shapes).

- p9. Out of curiosity: what types of calls are the calls by Jill in part C? Adding this might satisfy this reader's curiosity.

- p9, caption. "Some individuals are recorded": probably not too important, but is every individual always recorded either inside or outside (which is how the statement sounds)? Or are just some recordings done inside and some outside, with inidviduals both recorded inside and outside (as seems more logical, but not clear from the sentence)?

- p10. "A sketch of across-individual variability": "... in Bonono vocalizations"? Again, it is not entirely clear here whether this subsection is meant as a general technique/workflow or just as background for this specific dataset. An effort to increase clarity (here, but also in general in the first half of the article) would be good.

- p10. A lot more "F0"s.

- p10, 11. "like deterministic chaos": Please use a better acoustic term (something with "non-harmonic vocalisations" or "turbulent air flow" or so?)

- p10, figure 3. "PRAAT" was written "Praat" before.

- p10, figure 3. "50% or 80%": so these are centered around the median? So 50% is the inter-quartile range?

- p10, figure 3. When printed (at least, for me), the grey area is as good as invisible. I did not notice it until I read the caption.

- p11, 3-4. "while it is more central": "... more central in duration". "more central" sounds weird for these 2 unrelated dimensions.

- p11, figure 4. Again, the resolutin (or printing quality) is very low somehow, making the left plot almost useless. But even with better quality, it it still a very cluttered and overlapping plot, so any extra effort to reduce this is appreciated. However, the general idea is pretty good-looking!

- p11, figure 4, caption. "1/10th": does this mean 1 10th of the duration, 1 10th of the pitch range? Please shortly clarify. Or, if not important, leave out the 1/10th and just keep "miniature", which should be good enough for the figure.

- p11, 14. "MFCC": worth mentioning (and referencing) that MFCC originate and are often used in human speech processing.

- p12, 17. "derived from decision trees": How? Is there any way of intuitively specifying? The current phrasing poses more questions than it answers.

- p12, 17. "considered as one of the best methods". For what? Picking methods is always a tradeoff, depending on context etc. Please refrain from such big claims or put them better in context.

- p12, 23-27. Thank you for nicely pre-empting the obvious "but deep learning!" comments and explaining why it is not an option for SUNG datasets!

- p13, 1-2. "instead .... true/false positives and negatives": FP/FN/TP/TN are directly related to accuracy (being TP/(TP + FP)) and recall. So the claim that they would be better for imbalanced classes (implied by the part of the paragraph on p12) does not make sense to me.

- p13, 26. The main issue with MFCCs is not per se their number. I would argue that MFCCs (and cepstrums) in general are hard to intuitively interpret, independent of how many there are.

- p13, 19-34. Better link up the explanations of the 3 ensembling methods to their description. Already a "respectively" would do wonders, but by restructuring a bit more, it will become a lot clearer to the reader.

- p13, 33-39. Related to the major comment from above, but xgboost might be very confusing for the reader, as xgboost is presented as a basic, non-ensembled ML method. Please make sure to clarify this situation and avoid any confusion between the two contexts that "xgboost" might cause.

- p14, 1-7. Before, good arguments were made about keeping models simple (e.g., no deep networks). However, here, through ensembling, the model becomes suddenly a lot more complex. It would be good to discuss the link between data size and overfitting here, and discuss if ensembling would risk overfitting (and if not, why not?).

- p14, 9. What is the task? Please first introduce the goal (one sentence is fine) before diving into the results.

- p14, 11. What is the size of the training/test set split?

- p14, 22. "best performances ... with xgboost": however, in 2 of the 3 cases, SVMs outperform xgboost on the same feature set? At least, when I look at Figure 5. Please resolve.

- p14. When claiming MFCC is not working well, it is using way more than the typical number of filters. The typical number of MFCC coefficients I've seen being used is 13 (12 + energy), possibly + delta and deltadelta. These 39 features are still more than the 7 or 20 features of bioacoustics or DCT feature sets, but way less than the huge 192 used here. So, in order to convince me and other readers here that MFCCs are not appropriate, adding a feature set with a way smaller numbrer of coefficients would be good.

- p14, 30. "automatically extracted": it is not clear how DCT is more automatically extracted than the bioacoustics dataset. Both rely on having manual annotations and Praat's automatic pitch detection?

- p14, 32. "all seven configurations": which seven configurations? This is not described anywhere before this point and comes out of nowhere. (example of unclear order, as mentioned above)

- p14, 35. "acc" "bac": are these abbreviations used? They get introduced but not really used in the paragraphs after.

- p15, 1. This also goes for the stacked learners, so I don't get the "for the non-stacked learners" addition.

- p15, figure 5. It is not clear where the error bars originate. What is being repeated? (After reading further, I see that 100 samples are being taken, but this is not at all clear at this point in the manuscript.)

- p15, figure 5. It would be nice to have the chance level indicated with e.g. a dashed vertical line, for reference.

- p16, Table 2. Merge "config #" and "combined config" columns to not waste space. Just "configuration" should be clear enough

- p16, Table 2. A lot of nubmer here. I wonder if there's a better way (e.g. lightly coloring as some sort of heatmap) to make this more easily readable. I get lost with so many numbers to compare.

- p16, figure 6. Why? What is the interpretation of this figure? What do we learn from it? Please discuss the visualisation's insights.

- p17, figure 7. In principle, one could theoretically link up the features of the two sets. E.g. DCT2 would probably be linked to both f0 slopes. Potentially, a correlation between the two sets would be insightful.

- p17, 7. Significance is now mentioned way earlier than in Task 1. I would suggest to follow the same structure of pragraphs, as to have these two sections maximally comparable.

- p17, 8. "accuracies": balanced or unbalanced?

- p17, 8-9. What is the chance level?

- p18, 3. "positions" is a weird metaphor carying over from the plot. Is there a better way of putting this?

- p18. No discussion of DCT vs bioacoustics features? In this case, DCT doesn't seem to be quite as good, and this adds a caveat to the previous results of "DCT being almost as good as BA". So it is worth discussing this, just like in Task 1.

- p19, 1. "Adoption of strategies to mitigate it": Which are these?? Again, lots of detailed information was missing before, and this only gets clear after reading section VI. But really, the way imbalanced data is being dealt with cannot just be dropped here. Especially as it is one of the 4 letters of SUNG, the central theme of the manuscript. This needs to be much clearer and the structure and order of things deserves rethinking.

- p19, 1-3. An insightful conclusion drawn from the observed results! Very important, in the case of small data to note this!

- p20, figure 10. again, what are the error bars?

- p20, 6-10. Nice and clear introduction of the issue of data leakage!

- p20, 14. It was until now not clear at all that anything was being resampled! Again, this needs to be made a lot more clear, throughout the manuscript.

- p21, 17. "ignoring the constraint of predefined size for both sets": unclear to what is actually meant. Please clarify.

- p22, 2. "100 runs reported in the previous section": ??? I didn't know anything about 100 runs, in the previous section. I explicitly searched through the manuscript, and no 100 runs is being mentioned in section II!

- p22, 17. "..."

- p23, 3-4. It is unfortunate that it is called the "Skewed strategy". In the ideal case, one would not have to clarify to the reader that this worst case should not be attempted as a strategy in any way.

- p23, 6. What is BALI? Is there a reference or link or anything?

- p23, 11-12. I would go futher and say that the caveat is relevant in most cases. But yes, probably *even more* in the case of SUNG.

- p23-25. As mentioned above, it is absolutely unclear what data *exactly* UMAP runs on. This whole section needs to be clarified.

- p25, 4. "the average distance to data points ..." -> "the minimum average distance to data points ..."

- p27, 8-10. Surely, some sort of theoretical analysis should be able to link the DCT features to the BA measures (e.g., curvedness vs difference in start vs. end slope)? If you want to make a case of more research using DCT, this would be a useful extra discussion to convince people why DCT features are not just abstract numbers.

- p28, 16-23. Most of this text goes for UNsupervised methods and exploring groupings/clusters of data points. Please clarify how the situation is different in this manuscript, as you have labeled data and supervised methods.

- p28, 33. "adopting SVM": but in section II, claims are made that xgboost is better?

- p29, 35. Praat version number, for reproduceability?

- p30, 1. Praat was also used for harmonicity?

- p30, 12. "DCTHarm": This is the first time this is being used. Everywhere before it is just "DCT". What's the difference?

- p30, 19. "the second coefficient is directly proportional to the linear slope": I'm not sure if this is true, as it is not a linear slope (but rather half a period of a sine/cosine wave). It will capture the slope, yes, but it won't be directly proportional to a linear slope.

- p30, 32. "-peep-yelp)": typo

- p31, 1-4. Why? Please clarify why when wrong. Why was the one value "missing"? Did Praat detect it as "unvoiced"? And why reconstruct rather than just leave out the one data point? This whole paragraph causes a lot more questions than it solves.

- p32, 29. What train/test split %?

- p32, 31-32. I don't really see the importance of mentioning 25-thread parallelization to the current article, here. It should have no influence on the precision or other results, and kind of interrupts the other sentences.

- p35, 27. "glmnet"? Sounds like the name of an R library. But what is the full name of the model/algorithm?

- p35, 31. Is there anything on BALI for readers to check, or to help reproduce this research? Any sort of reference or other information? Otherwise, this is really hard to reproduce the results.

- p36, 16-24. This whole part is rather unclear. It is not clear whether the 100 iterations are used as separate datapoints or as features of one data point or ... Please rewrite and clarify, potentially with an example of the sort of data you end up before UMAP.

p37, 13-18. In general, I don't see the point here (where no time performance is reported either way) to specify the full computational hardware and threading. But if the authors insist, why not.

Reviewer #2: The manuscript tackles a quite common issue when analysing animal communication and vocal repertoires: datasets constituted by a small amount of calls, produced by few individuals, recorded in suboptimal conditions. The authors relied on the analysis of information encoding in bonobo repertoire, improving our understanding, to propose a methodological, operational workflow that can be used by researchers dealing with similar assessments. The analyses conducted employed different kinds of feature extraction, state-of-the-art classification techniques, supported by a thorough literature review, adopting two evaluation tasks (the individual identity and the call type identification). The results support the conclusions the authors drew and the helpful recommendations they suggested. The authors indeed found how adopting multiple feature sets may be good practice for extracting non-redundant information, proposed a generalizable practical approach that can be helpful when dealing with similar problems and recommendations on how adequately treat small and unbalanced datasets where calls occur in sequences, which can be true for many animal species. Methodology is broadly and accurately described, datasets and analysis codes are available and deposited in an appropriate and accessible repository, allowing to reproduce the study. Concluding, the work is scientifically sound, well constructed and presented. However, the different feature extraction, classification techniques, and evaluation tasks, lead to a workflow which can be a little hard to follow: I recommend to indicate what methods, how many call types, individuals, and sets are used in which task (for instance modifying figure 1 illustrating the workflow accordingly; see comment to P14-L32).

Below punctual comments.

ABSTRACT

L7 For clarity, I suggest to add why 'SUNG datasets offer a distorted vision of communication systems'

INTRODUCTION

P2

L6 I would rethink 'success stories'

LL7-9 This sentence needs to be supported by a reference

L13 Expensive in terms of?

L14 Are they really invaluable though?

LL23-26 Is there a reference that can support this statement?

L28 Is there a reference regarding the 'playback experiments' that can be cited?

LL29-34 Please, rephrase this sentence

P3

L13 Is the pDFA still the standard approach in bioacoustic analyses though?

LL25-34 Nice! See also the findings of Sainburg et al. 2020 (10.1371/journal.pcbi.1008228) for a comparison among those methods.

P4

L17-18 'more accurate' and 'more comprehensive understanding' compared to?

L20 Which are the 'difficulties inherent in raw naturalistic dataset'?

LL27-28 This will be probably stated afterwards but how did you carefully choose an adequate performance metrics? What makes a performance metrics adequate?

P6

L16 What do you mean by 'ubiquitous'?

L23 From figure 2 is difficult to understand which are the ten call types

L26-27 '...tonal contour (general shape and duration)': how is the general shape quantified?

P7

L5 maybe 'corpora' instead of 'corpus'?

L16-18 I am not sure I understood this sentence

L29 'whether an acoustic feature is relevant' in discriminating the emitter's identity?

L37 '20 adult' how many males and females?

P8

L3 'vocalizations.. double-checked by two other experimenters': did you measure the inter-raters reliability? If so, how?

L3 I believe the correct reference is Boersma and Weenink

P10

L3 'the whole corpus' excluding grunts and screams?

L4 Which kind of smoothing has been employed?

L4 Maybe 'coarse' instead of 'gross'?

L12 How has the 'harmonicity' been measured?

P11

LL12-17 To increase the understandability, I would add which parameters the Bioacustic, MFCC, and DCT set consider, respectively.

P12

L1 Can you expand the extraction of acoustic features section? How are the tonal contour (previous page) and the acoustic roughness measured?

P14

L21 So MFCC are used to identify call types but not the individual identity?

LL27-28 Can you indicate some references here?

L32 Which are the 'seven configurations'? I got it when reading table 2 but there are a lot of analyses and different sets employed for the different tasks. The readers will be probably benefit from a clear explanation of what methods, call types, and sets are used in which cases (for instance modifying figure 1 illustrating the workflow accordingly could be enough).

P20

L3 'confirming the multidimensional and complex nature of the individual vocal signature' is more a discussion.

L7 which is the relationship between these vocal sequences and the call types indicated in the previous paragraphs?

P21

L8-12 The third strategy represents an interesting approach!

Fig 11: So, if I got it right, is the influence of data leakage measured when classifying individual signatures only (besides the subset of three-call sequences)?

RESULTS

P21

LL19-20 Cool result!

P22

LL14-15 '259 sequences consist of only one call, 111 consist of 2 calls': calls refers to the number of calls or to the call types? Do the sequences include more than one single call type?

p23

LL1-3 Interesting result

L6 'Tools such as BALI': link this to the relative paragraph (an in-text citation like see paragraph 'Evaluating data leakage / BaLi' is enough)

p26

LL8-9 Average silhouette values of 0.9 and 0.96 for call types and individuals identity, respectively. With regards to figures 13 and 14 how are these values averaged and measured?

LL13-15 Indeed

P27

L1 This sentence needs a reference

LL1-2 Can you expand this?

L7 Did your set include the same call type used in Keenan et al. 2016?

L31 interesting

P29

L24 Maybe something like 'tackling' instead of 'grasping'?

P32

L22 Can you add a reference here?

LL27-28 What does 'some observations are inherently more difficult to classify than other' mean?

P34

shouldn't the whole paragraph 'Tuning procedure' report at least some references supporting the choices that have been made?

P36

L8 Why using 15 neighbors if the analyses either 10 individuals or 5 call types?

Reviewer #3: I have reviewed the paper by Arnaud et al (PCOMPBIOL-D-22-00956).

I read the paper from Arnaud et al with great interest and applaud the authors for attempting to improve and accelerate the methodological tools available for the analysis and classification of animal vocalisations.

In principle I think this paper falls nicely within the remit of PLoS computational biology. I do still have a few suggestions and questions that it would be great to get feedback from the authors on.

1. My biggest issue is that I struggled a little to see the bigger picture with this paper; specifically, what precisely are the authors interested in achieving with their analysis of bonobo vocalisations? From what I can tell the goals range from classifying call types, assessing the influence of sample size and noise on such classification, comparing different methods/approaches to assessing the shortcomings of previous core approaches in the field (i.e. pDFA) and investigating biological questions associated with the encoding of information sets in animal vocalisations. The reason I bring this up is because I worry that by trying to address so much in one single paper the authors risk readers being overwhelmed and the core message and advances of the paper being lost. If possible, I would therefore propose that the authors try to condense their goals perhaps to the ones that are most pressing and most relevant for the field at the moment. I would argue that even a comparison of different methods to resolve the different call types (i.e. repertoires) would already be incredibly informative and ultimately easier to digest for the field. Undertaking this with SUNG datasets adds the additional novel component that is so relevant to many researchers working with animal communication and really elevates the paper and its contribution. Highlighting the shortcomings of other approaches and attempting to capture additional variation as a result of phenotypic differences, I think, could be addressed in a follow up study or eluded to in the discussion as necessary future work.

2. I think a clearer reasoning or justification for using bonobo vocal data would also be valuable. Perhaps mention that, of the primates, great ape vocal data is notoriously difficult to collect and is riddled with issues resulting from sample size and background noise etc. Monkey and non-primate data sets suffer from these issues less because there are often more groups from which data can be collected whereas with apes, researchers are often restricted to working with one group leading to limited subjects and ultimately constrained sub-optimal data sets.

3. I think it would be useful to provide spectrograms of the main 5 call types the researchers focus on for their analyses.

4. Since the bonobo repertoire is rather tonal in its nature, as far as I can see, the authors are not confronted with comparing voiced and unvoiced calls. Since this is not always the case in many animal call repertoires, some guidance with regards to the analysis of vocal repertoires comprising both voiced and unvoiced calls would be useful. In my experience the absence of an F0 can massively complicate call classification and comparisons and a suitable solution (other than relying on less comprehensive non-F0 based measures or features) is not available.

**Have the authors made all data and (if applicable) computational code underlying the findings in their manuscript fully available?**

Reviewer #1: **No: **The data has been made available (and is the most important part), but no code has been attached.

However, I leave this up to the editor on whether this is necessary. If the right methods and version numbers are added, this would also be fine with me.

Reviewer #2: Yes

Reviewer #3: Yes

PLOS authors have the option to publish the peer review history of their article (what does this mean?). If published, this will include your full peer review and any attached files.

Reviewer #1: No

Reviewer #2: No

Reviewer #3: No

Figure Files:

Data Requirements:

Please note that, as a condition of publication, PLOS' data policy requires that you make available all data used to draw the conclusions outlined in your manuscript. Data must be deposited in an appropriate repository, included within the body of the manuscript, or uploaded as supporting information. This includes all numerical values that were used to generate graphs, histograms etc.. For an example in PLOS Biology see here: http://www.plosbiology.org/article/info:doi%2F10.1371%2Fjournal.pbio.1001908#s5.
---

## [Editor Report · Decision Letter 1]

10 Nov 2022

Dear Dr. Coupé,

Thank you very much for submitting your manuscript "Improving the workflow to crack Small, Unbalanced, Noisy, but Genuine (SUNG) datasets in bioacoustics: the case of bonobo calls" for consideration at PLOS Computational Biology. As with all papers reviewed by the journal, your manuscript was reviewed by members of the editorial board and by several independent reviewers. The reviewers appreciated the attention to an important topic. Based on the reviews, we are likely to accept this manuscript for publication, providing that you modify the manuscript according to the review recommendations.

Dear Vincent and co-authors,

Thank you for addressing the comments of the three reviewers on the first round of revision for your paper. Since you have addressed these relatively well, I will not send it back to them. On the other hand, I have my own set of comments that I would like you to address before publication. Bioacousticians working with SUNG data sets will certainly read this piece with interest and I am looking forward to your reply.

Best wishes,

Frederic Theunissen

1. I also appreciate the very significant contribution of the permuted-DFA, but in your introduction it sounds like it is a concept/approach that was originally developed by Mundry and Sommer. Their contribution was essential in bioacoustics but the concepts of non-independence of data (think random effects in mixed-effect modeling), permutation tests and cross-validation were well described before that in the statistical literature (and prior to the data leakage formulation used in machine learning – checkout the wikipedia page for permutation test for example – it will bring you to the 1930…). I would introduce the pDFA (and the contribution of Mundry and Sommer to its application in bioacoustics) with that more inclusive historical/statistical perspective.

2. Along the same lines, in the rest of the paper, you keep on using pDFA but in fact, all your methods use a form on permutation test by performing the right type of cross validation (see below). The difference between the classifiers are a linear discriminant analysis (DFA) versus non-linear methods that find other boundaries between classes (including potentially multimodal/donuts in case of xgboost). Thus, first, I would suggest – using DFA vs other classifiers instead of pDFA vs other classifiers in the rest of the manuscript. Second, in your discussion, you mention things like “Because it may not be sensitive enough, pDFA…” ( p. 32) when you compare the DFA to other classifiers Saying “not sensitive enough” is quite vague. I encourage you to do better and explain the differences between classifiers both more explicitly and more didactically. For example, I suggest you try to generate a figure with the different decision boundaries that are obtained with the different classifiers.

3. Figure 1 A. I know that this is a workflow figure but the cartoon for the spectro-temporal templates is a bit cryptic. I know it becomes clear as one reads the paper but why not add x and y axis labels? .Also note that you illustrate with only 4 templates for call types (and not 10) and in the individual plot maybe you should have an individual grouping color (otherwise it looks like more than 20 individuals)?

4. P. 10 “Vocalizations were manually segmented, identified and then double-checked by two other experimenters (through a consensus decision-making) with Praat (Boersma & Weenink, 2022), based on the visual inspection of signals on spectrograms.” This sentence is hard to parse - I suggest “Vocalizations were manually segmented, identified and then double-checked by two other experimenters (through a consensus decision-making) based on a visual inspection of the signals on spectrograms and an estimation of the fundamental frequency, fo, using the speech analysis software Praat (citation).” This will also help with the sentence that starts with “The audio quality of the recordings was variable…” where you mention fo and Praat again without defining Fo.

5. P. 18. Define AUC once as the Area Under the ROC Curve.

6. Fig. 7 The font size in this confusion matrix is too small. Use all the space you have. This comment is valid for all your figures. Please take a good look at your font sizes in all the figures and make then legible and more appealing.

7. There is no specific reason that prevents you from using DFA with a high dimensional feature spaces any more so that with other classifiers. In fact, one could argue that it is the opposite. DFA is a simpler model than SVN, NN, or tree classifiers in the sense that there are fewer parameters to fit per feature dimension. Just as there are regularization parameters (hyperparameters) for the other classifiers, you could regularize the DFA to prevent overfitting – for example by performing a PCA on your feature space and preserving a variable number of PCs. This could be used on the MFCC feature space for example (see Elie and Theunissen, 2016). A regularized DFA might also boost your performance for the other feature spaces.

8. In your description of the permuted DFA (p.37), it was unclear to me whether the permutation was performed within each group along the dimension not being tested or across all groups. For example, if you are testing for call-type, one should preserve the identity information and permute the call-type label within each individual but not across all individuals. And vice-versa for testing for identity (Note this is what was done in Elie and Theunissen, 2016 for call types – read cross-validation section on p. 294 - and Elie and Theunissen, 2018 for caller id). Did you do this? If so, please be more explicit about it in the methods. If not, I suggest that you perform the permutation using this more rigorous approach. It deals with the unbalanced data (or, equivalently, with the random effect of id when one analyzes call-type).

9. Related to point 7 but for generating a fair testing set: I appreciate the fact that you use Bali to generate testing sets with specific objectives that control for random effects (such as two sounds coming from the same sequence). Just as in the permuted DFA, a strong approach is to generate a testing set that deals with the lack of independence in individual samples (also known as a random effect creating data leakage). For call-type, the testing sets could only have individuals that are not in training set. For id, the testing set could have call types that are not in the training set (or you should perform an analysis per call type). For the sequence effect, the sequences need to be all in the training set or all in the testing set, etc. If you do this, you will clearly get the most conservative estimate. And I am not completely sure what one would gain by applying the genetic algorithm. As far as I can tell you need a genetic algorithm because of you are getting the best solution for a fixed sized testing set (please correct me if I don’t understand – and if so make sure to also state more explicitly the reason for using Bali in the manuscript). Thus, you might have to generate testing sets of different sizes (for example as you include and exclude different individuals for assessing call type discrimination). But I don’t think this is a problem – you can always to the correct weighting to get the average performance.

Sincerely,

Frédéric E. Theunissen

Academic Editor

PLOS Computational Biology

Natalia Komarova

Section Editor

PLOS Computational Biology

Dear Vincent and co-authors,

Thank you for addressing the comments of the three reviewers on the first round of revision for your paper. Since you have addressed these relatively well, I will not send it back to them. On the other hand, I have my own set of comments that I would like you to address before publication. Bioacousticians working with SUNG data sets will certainly read this piece with interest and I am looking forward to your reply.

Best wishes,

Frederic Theunissen

1. I also appreciate the very significant contribution of the permuted-DFA, but in your introduction it sounds like it is a concept/approach that was originally developed by Mundry and Sommer. Their contribution was essential in bioacoustics but the concepts of non-independence of data (think random effects in mixed-effect modeling), permutation tests and cross-validation were well described before that in the statistical literature (and prior to the data leakage formulation used in machine learning – checkout the wikipedia page for permutation test for example – it will bring you to the 1930…). I would introduce the pDFA (and the contribution of Mundry and Sommer to its application in bioacoustics) with that more inclusive historical/statistical perspective.

2. Along the same lines, in the rest of the paper, you keep on using pDFA but in fact, all your methods use a form on permutation test by performing the right type of cross validation (see below). The difference between the classifiers are a linear discriminant analysis (DFA) versus non-linear methods that find other boundaries between classes (including potentially multimodal/donuts in case of xgboost). Thus, first, I would suggest – using DFA vs other classifiers instead of pDFA vs other classifiers in the rest of the manuscript. Second, in your discussion, you mention things like “Because it may not be sensitive enough, pDFA…” ( p. 32) when you compare the DFA to other classifiers Saying “not sensitive enough” is quite vague. I encourage you to do better and explain the differences between classifiers both more explicitly and more didactically. For example, I suggest you try to generate a figure with the different decision boundaries that are obtained with the different classifiers.

3. Figure 1 A. I know that this is a workflow figure but the cartoon for the spectro-temporal templates is a bit cryptic. I know it becomes clear as one reads the paper but why not add x and y axis labels? .Also note that you illustrate with only 4 templates for call types (and not 10) and in the individual plot maybe you should have an individual grouping color (otherwise it looks like more than 20 individuals)?

4. P. 10 “Vocalizations were manually segmented, identified and then double-checked by two other experimenters (through a consensus decision-making) with Praat (Boersma & Weenink, 2022), based on the visual inspection of signals on spectrograms.” This sentence is hard to parse - I suggest “Vocalizations were manually segmented, identified and then double-checked by two other experimenters (through a consensus decision-making) based on a visual inspection of the signals on spectrograms and an estimation of the fundamental frequency, fo, using the speech analysis software Praat (citation).” This will also help with the sentence that starts with “The audio quality of the recordings was variable…” where you mention fo and Praat again without defining Fo.

5. P. 18. Define AUC once as the Area Under the ROC Curve.

6. Fig. 7 The font size in this confusion matrix is too small. Use all the space you have. This comment is valid for all your figures. Please take a good look at your font sizes in all the figures and make then legible and more appealing.

7. There is no specific reason that prevents you from using DFA with a high dimensional feature spaces any more so that with other classifiers. In fact, one could argue that it is the opposite. DFA is a simpler model than SVN, NN, or tree classifiers in the sense that there are fewer parameters to fit per feature dimension. Just as there are regularization parameters (hyperparameters) for the other classifiers, you could regularize the DFA to prevent overfitting – for example by performing a PCA on your feature space and preserving a variable number of PCs. This could be used on the MFCC feature space for example (see Elie and Theunissen, 2016). A regularized DFA might also boost your performance for the other feature spaces.

8. In your description of the permuted DFA (p.37), it was unclear to me whether the permutation was performed within each group along the dimension not being tested or across all groups. For example, if you are testing for call-type, one should preserve the identity information and permute the call-type label within each individual but not across all individuals. And vice-versa for testing for identity (Note this is what was done in Elie and Theunissen, 2016 for call types – read cross-validation section on p. 294 - and Elie and Theunissen, 2018 for caller id). Did you do this? If so, please be more explicit about it in the methods. If not, I suggest that you perform the permutation using this more rigorous approach. It deals with the unbalanced data (or, equivalently, with the random effect of id when one analyzes call-type).

9. Related to point 7 but for generating a fair testing set: I appreciate the fact that you use Bali to generate testing sets with specific objectives that control for random effects (such as two sounds coming from the same sequence). Just as in the permuted DFA, a strong approach is to generate a testing set that deals with the lack of independence in individual samples (also known as a random effect creating data leakage). For call-type, the testing sets could only have individuals that are not in training set. For id, the testing set could have call types that are not in the training set (or you should perform an analysis per call type). For the sequence effect, the sequences need to be all in the training set or all in the testing set, etc. If you do this, you will clearly get the most conservative estimate. And I am not completely sure what one would gain by applying the genetic algorithm. As far as I can tell you need a genetic algorithm because of you are getting the best solution for a fixed sized testing set (please correct me if I don’t understand – and if so make sure to also state more explicitly the reason for using Bali in the manuscript). Thus, you might have to generate testing sets of different sizes (for example as you include and exclude different individuals for assessing call type discrimination). But I don’t think this is a problem – you can always to the correct weighting to get the average performance.

Figure Files:

Data Requirements:

Please note that, as a condition of publication, PLOS' data policy requires that you make available all data used to draw the conclusions outlined in your manuscript. Data must be deposited in an appropriate repository, included within the body of the manuscript, or uploaded as supporting information. This includes all numerical values that were used to generate graphs, histograms etc.. For an example in PLOS Biology see here: http://www.plosbiology.org/article/info:doi%2F10.1371%2Fjournal.pbio.1001908#s5.

Reproducibility:

References:

---

## [Editor Report · Decision Letter 2]

1 Mar 2023

Dear Dr. Coupé,

We are pleased to inform you that your manuscript 'Improving the workflow to crack Small, Unbalanced, Noisy, but Genuine (SUNG) datasets in bioacoustics: the case of bonobo calls' has been provisionally accepted for publication in PLOS Computational Biology.

Best regards,

Frédéric E. Theunissen

Academic Editor

PLOS Computational Biology

Natalia Komarova

Section Editor

PLOS Computational Biology

Dear Vincent, Christophe and co-authors,

Thank for a careful revision of your manuscript. I believe that the paper is now much clearer - and clearly very complete. Congratulations. I have a few minor recommendations for the final submission.

Best wishes,

Frederic.

Minor

1. “We generated S-UMAP and silhouette scores for the description of the individual signatures and the call types with the combined features of the three Bioacoustic, DCT and MFCC sets (217 features in total) and…”. I think it would useful to spell out how you get to 217, meaning how it is divided among Bio, DCT and MFCC. (I see that it is on p.33 in the Main findings but it could be specified well above).

2. P15-16 Why not attempt SE on silhouette scores? For example, from bootstrapping with cross validation? (or the other techniques you describe further down in the results).

3. “Obviously, bioacousticians are not necessarily obsessed with achieving the best classification performance. In fact, a trade-off between performance and computational time should in principle be sought.” I would add that the trade-off also includes interpretation (or mechanistic explanation).

---

## [Editor Report · Acceptance letter]

6 Apr 2023

PCOMPBIOL-D-22-00956R2 

Improving the workflow to crack Small, Unbalanced, Noisy, but Genuine (SUNG) datasets in bioacoustics: the case of bonobo calls

Dear Dr Coupé,

I am pleased to inform you that your manuscript has been formally accepted for publication in PLOS Computational Biology. Your manuscript is now with our production department and you will be notified of the publication date in due course.

With kind regards,

Anita Estes
